# The molecular basis for recognition of 5′-NNNCC-3′ PAM and its methylation state by *Acidothermus cellulolyticus* Cas9

Anuska Das[1], Travis H. Hand[1], Chardasia L. Smith[1], Ethan Wickline[2], Michael Zawrotny [1] & Hong Li[1,2✉]

*Acidothermus cellulolyticus* CRISPR-Cas9 (AceCas9) is a thermophilic Type II-C enzyme that has potential genome editing applications in extreme environments. It cleaves DNA with a 5′-NNNCC-3′ Protospacer Adjacent Motif (PAM) and is sensitive to its methylation status. To understand the molecular basis for the high specificity of AceCas9 for its PAM, we determined two crystal structures of AceCas9 lacking its HNH domain (AceCas9-ΔHNH) bound with a single guide RNA and DNA substrates, one with the correct and the other with an incorrect PAM. Three residues, Glu1044, Arg1088, Arg1091, form an intricate hydrogen bond network with the first cytosine and the two opposing guanine nucleotides to confer specificity. Methylation of the first but not the second cytosine base abolishes AceCas9 activity, consistent with the observed PAM recognition pattern. The high sensitivity of AceCas9 to the modified cytosine makes it a potential device for detecting epigenomic changes in genomes.

[1] Institute of Molecular Biophysics, Florida State University, Tallahassee, FL 32306, USA. [2] Department of Chemistry and Biochemistry, Florida State University, Tallahassee, FL 32306, USA. ✉email: hong.li@fsu.edu

CRISPR (clustered regularly interspaced short palindromic repeats)-associated protein 9, CRISPR-Cas9, found in bacteria, has been repurposed for many gene editing and diagnostic applications[1–6] due to the ease with which it can be programmed to target virtually any DNA sequence. After the initial proof of utility in human cells[7–10], efforts have been made to further improve and adapt Cas9 for a broader range of applications, many of which benefited from the emerging molecular mechanisms of Cas9[11–15]. A functional Cas9 is comprised of the Cas9 protein itself and two partner RNAs, a CRISPR RNA (crRNA) and a trans-activating RNA (tracrRNA). The crRNA contains the antisense region (spacer) complementary to the target DNA (protospacer) whereas the tracrRNA, which pairs with the repeat region of the crRNA, contains a multi-stem scaffold that is bound to Cas9. The crRNA and tracrRNA can be covalently linked through a tetraloop into a single-guide RNA (sgRNA) without significant loss of activity.

A critical element of the DNA target recognized by Cas9 is called protospacer adjacent motif (PAM). PAM is typically 2–8 nucleotides long and located immediately downstream of the protospacer. Cas9 binds to PAM, which causes local unwinding of the protospacer so it can pair with the spacer in crRNA. Each Cas9 is specific for a unique PAM sequence, although many characterized Cas9 accommodate substitutions of similar nucleotides in the PAM. The widely used *Streptococcus pyogenes* Cas9 (SpyCas9) is the most active on targets with the 5′-NGG-3′ but weakly on those with the 5′-NAG-3′ PAM[16,17]. SpyCas9 seems to be insensitive to DNA modifications within both the protospacer and the PAM ($^{5m}$CGG, for instance) regions[16,18]. Several other Cas9s that are smaller in size and thus better for in vivo delivery recognize longer and purine-rich PAMs[19,20]. Whereas promiscuous PAM sequences lead to a greater targeting range, they are more inducive to potential off-targets, which could hinder safe therapeutic application of Cas9. Strict PAM sequences, on the other hand, theoretically impart greater specificity but narrow the range of target selection. Specific Cas9–PAM interactions have also been exploited for enrichment and detection of the low-level tumor variants in cell-free DNA[21]. Given the essentiality of the PAM, the molecular basis for its interaction with Cas9 will thus benefit further development of Cas9-based technology.

Structural studies of Cas9–sgRNA–DNA complexes from seven species[22–28], covering all known Type II subtypes (Type II-A, II-B, and II-C), have illustrated detailed molecular interactions between Cas9 and their respective DNA substrates, especially the PAM sequences. Invariably, all Cas9 use charged or polar amino acids to form a network of hydrogen bonds with the PAM nucleotides directly. Arginine, and to a lesser degree, asparagine, are the most frequently observed PAM-interacting residues. Type II-A and II-B Cas9s primarily contact PAM bases on the nontarget strand, whereas Type II-C Cas9s make contacts with PAM bases on both the target and the nontarget strands. Interestingly, the frequently observed PAM bases contacted by Cas9s are purines (A and G), perhaps due to their rich hydrogen bonding sites. Despite the common features observed in the Cas9–PAM interactions, each Cas9 is unique in its method to confer specificity.

We previously reconstituted and characterized Cas9 from *Acidothermus cellulolyticus* (AceCas9)[29]. AceCas9 is small in size (1138 amino acids) and recognizes a 5′-NNNCC-3′ (where N is any nucleotide) PAM[29]. Due to its thermostability, AceCas9 has been used for gene editing in model thermophilic bacteria for biofuel processing[30]. Here we further characterized the PAM recognized by AceCas9 and report crystal structures of AceCas9 without its HNH domain (AceCas9–ΔHNH) bound with a guide RNA and DNA targets. We show that AceCas9 contacts the 5′-NNNCC-3′ PAM by forming a network of hydrogen bonds

with both the cytosine and guanine bases. Mutation of the 5′-NNNCC-3′ to 5′-NNNTC-3′ severely impaired AceCas9 activity and destabilized the AceCas9–PAM interaction. Strikingly, we discovered that AceCas9 is sensitive to the methylation state of the first but not the second cytosine base of the PAM, consistent with the observed mode of interactions.

## Results

**AceCas9 requires 5′-NNNCC-3′ PAM for efficient cleavage**. Our previous work showed that 5′-NNNCC-3′ is a functional PAM for AceCas9 but did not explore other possible PAMs that may also support AceCas9 functions[29]. To more broadly identify PAM for AceCas9, we applied a plasmid depletion coupled with next-generation sequencing (NGS) method. A plasmid library containing a cognate protospacer followed by seven randomized base pairs was subjected to cleavage by either a functional AceCas9 or AceCas9 without its guide RNA. The uncleaved products from both reactions were sequenced by NGS and compared to reveal the preferred PAM sequences for AceCas9 (Fig. 1a). Not surprisingly, 5′-NNNCCNN-3′ was the only PAM significantly depleted from the plasmid library by AceCas9 (Fig. 1a and Supplementary Data File 1). The next frequently depleted sequence motif (5′-NNNACNN-3′) had a computed PAM depletion score significantly higher than that of 5′-NNNCCNN-3′, suggesting that it is a weakly recognized PAM by AceCas9 (Fig. 1a and Supplementary Data File 1). We constructed eight plasmid DNA substrates containing various PAM sequences and subjected them to in vitro cleavage by AceCa9 both in their supercoil and pre-linearized forms. Incubation of molar excess AceCas9 with the DNA substrates at 50 °C for one hour resulted in complete double-stranded cleavage of both the supercoiled and pre-linearized 5′-NNNCCNN-3′-containing substrates but only supercoiled 5′-NNNACNN-3′-containing substrate (Fig. 1b). The pre-linearized form lacks the favorable energy stored in the negative supercoil and would be cleaved by AceCas9 at a slower rate[29]. Consistent with the weak cleavage in vitro, the 5′-NNNACNN-3′-containing substrate also failed to be cleaved in vivo to allow bacterial growth in an antibiotic selection assay (Fig. S1). These results also showed that AceCas9 has no preference for the first three nucleotides (Fig. 1b). The 5′-NNNCCNN-3′ PAM is thus the most specific form recognized by AceCas9.

**AceCas9 is sensitive to DNA methylation**. We have shown that mutations of the two cytosine bases are detrimental to AceCas9 function (Fig. 1b)[29]. To specifically distinguish between the target and the nontarget strand recognized by AceCas9, we constructed DNA targets with the 5′-NNNCC-3′ PAM where C4* or C5* or both is substituted by 5-methyl-cytosine ($^{5m}$C*). Methylation of C4* or C5* on the nontarget strand alone maintains the integrity of its base-paired guanosine, G(-4) or G(-5), on the target strand. We subjected the methylated DNA targets to cleavage by AceCas9 in vitro and in vivo, where possible, by the previously established cell surviving assay[31].

We first tested AceCas9 activity on methylated synthetic DNA oligo substrates. The target strand labeled with a 5′-fluorescent tag was annealed with a nontarget strand bearing no methylation (WT), $^{5m}$C4*, $^{5m}$C5*, or $^{5m}$C4*$^{5m}$C5* to form the double-stranded DNA targets for AceCas9. The wild-type and the $^{5m}$C5* targets were cleaved completely (Fig. 1c). However, mutation of C5* to any other nucleotides prevented cleavage from AceCas9 (Fig. 1b)[29], supporting that AceCas9 recognizes this base pair through its contacts with G(-5) that bases pairs with C5*. In contrast, $^{5m}$C4* or $^{5m}$C4*$^{5m}$C5* completely blocked cleavage by AceCas9 (Fig. 1c), suggesting a specific interaction between AceCas9 and C4*.

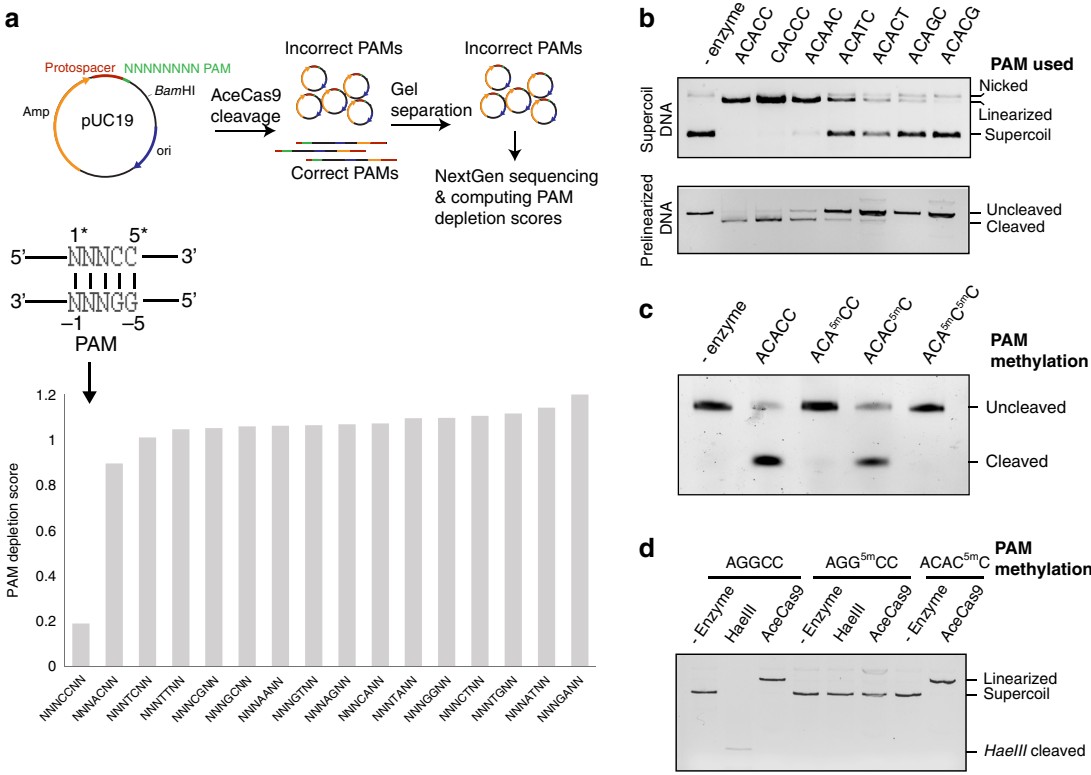

**Fig. 1 Determination of the protospacer adjacent motif (PAM) for AceCas9. a** The plasmid depletion assay scheme and the result. Seven base pairs immediately downstream of the protospacer are randomized. The plasmid depletion score is plotted for all possible dinucleotides at positions 4* and 5* and those of all other combinations are listed in Supplementary Data File 1. The PAM depletion score was computed as the ratio of the read frequencies in the AceCas9-treated and the untreated samples obtained from the Illumina sequencing (see details in "Methods"). A low score identifies the PAM. **b** DNA cleavage activities of AceCas9 on plasmid substrates with various PAM sequences that differ in positions 1*–5*. Each DNA substrate at 6 nM concentration was incubated with 1 μM AceCas9 at 50 °C for 1 h followed by gel separation and ethidium bromide staining. Top: Cleavage results of supercoil plasmids. Bottom: Cleavage results of pre-linearized plasmids. **c** DNA cleavage activities of AceCas9 on DNA oligomers containing either non-methylated or methylated PAM. The target DNA strand is not methylated and contains a 5′-hexachlorofluorescein (HEX) tag for visualization. The nontarget strand contains methylation on C4*, C5*, or both, respectively, and was heat-annealed with the target strand at 1 nM concentration before incubating with 1 μM AceCas9 RNP for 50 min at 50 °C. **d** DNA cleavage activities of AceCas9 on DNA plasmids containing non-methylated and or methylated PAM. Methylation of C4* was achieved by treatment of the DNA plasmid containing the PAM followed by GGCC sequence with *HaeIII* methyltransferase. The *HaeIII* restriction endonuclease (*HaeIII*) was used to verify the methylation. Methylation of C5* was achieved in a *dcm1*-positive *E. coli* by amplification of the plasmid containing the CCWGG (W = A or T) sequence. Source data are provided as a Source data file.

We then tested AceCas9 activity on methylated plasmid substrates. We used *HaeIII* methyltransferase to specifically methylate C4* or *dcm1* to methylate C5*. Methylation at the respective positions was verified by *HaeIII* restriction digestion or bisulfite sequencing (Fig. 1d and Fig. S2a). Similar to the result with DNA oligo substrates, 5mC4*-containing PAM was deleterious while 5mC5*-containing PAM had no effect on AceCas9 activity (Fig. 1d). The inactivity of the 5mC4*-containing PAM DNA with AceCas9 was not due to its inability to bind AceCas9 (Fig. S2b) and may thus likely due to a defective unwinding process.

To test the in vivo AceCas9 activity on DNA containing 5mC5*, we transformed the *dcm1*-methylated plasmid baring the *ccdb* toxic gene into BW25141 *E. coli* cells and performed the survival assay. Transformation of the plasmid encoding AceCas9 and its sgRNA into the cell harboring the 5mC5*-containing target plasmid resulted in full survival on arabinose plate similar to the result obtained with the target plasmid without methylation (Fig. S2c), suggesting that AceCas9 is insensitive to methylation on C5* position, again suggesting a lack of interaction with AceCas9.

**Crystal structure of an AceCas9–sgRNA–DNA complex**. We solved a crystal structure of the truncated AceCas9 that lacks its HNH catalytic domains (AceCas9–ΔHNH) (Fig. 2a). As expected, AceCas9–ΔHNH nicks the plasmid DNA substrate due to the absence of the HNH but the presence of the RuvC catalytic domain (Fig. 2a). To form AceCas9–ΔHNH–RNA–DNA crystals, we explored several sgRNA and target DNA oligos with varying length and sequences (Fig. 2b). A truncated but active guide RNA, sgRNA94, a target strand (TS) DNA comprised of the 20-nt complementary and a 10-nt PAM region, and a nontarget strand (NTS) DNA that base pairs with the PAM region of the TS (Fig. 2b) allowed cocrystallization with AceCas9–ΔHNH. The structure of the ternary complex was solved by a combination of molecular replacement and single-wavelength anomalous diffraction methods and refined at 2.9 Å to crystallographic residual factors of 0.22/0.26 (R/Rfree) ("Methods", Table S1). RNA and DNA nucleotides could be placed into the density unambiguously (Fig. 2b). Tracing of AceCas9 amino acids was assisted by matching the difference anomalous Fourier peaks with the six methionine residues. In the final model, we were able to place 949 of the total 988 protein residues, 91 of the 94 sgRNA94

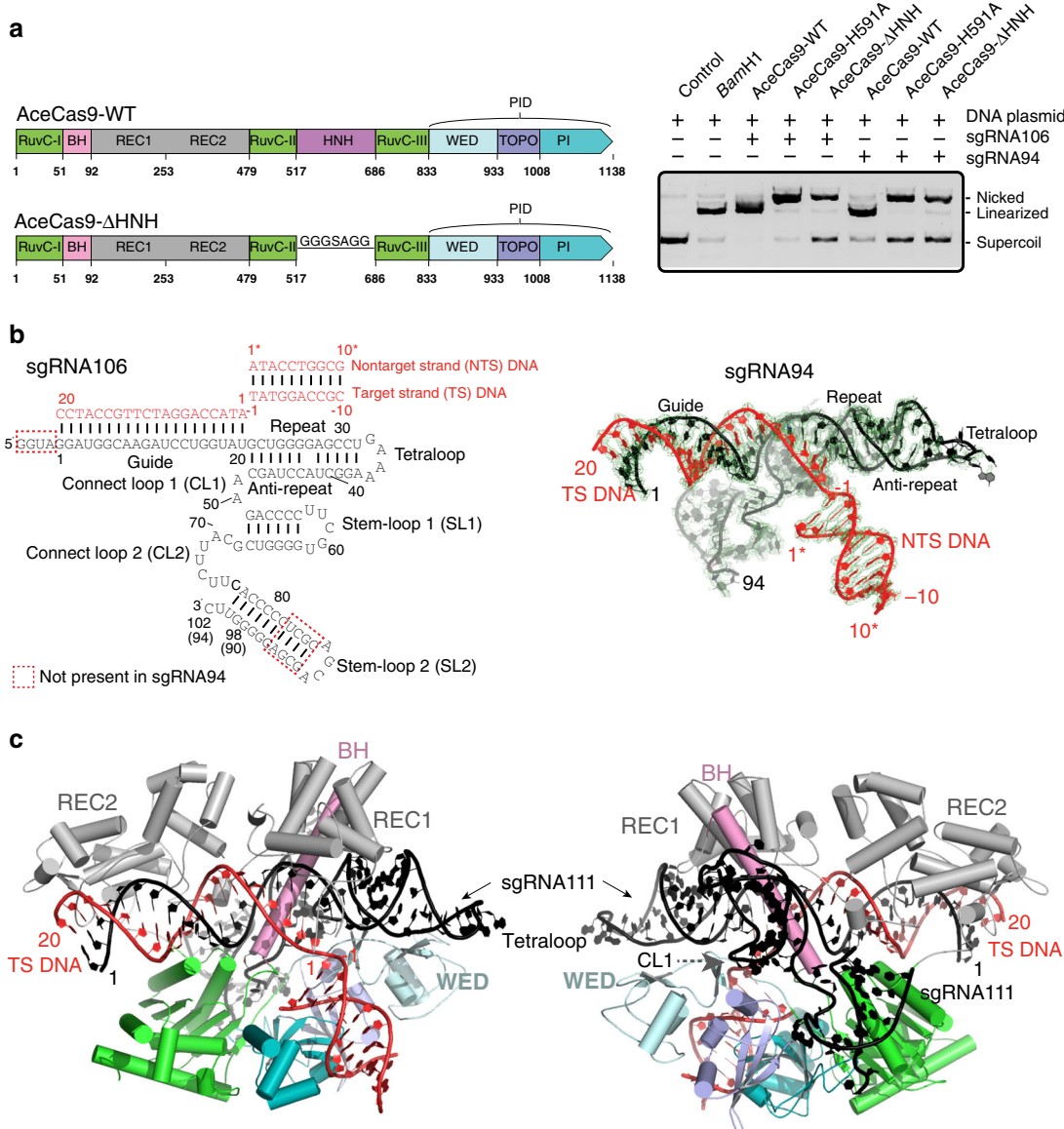

**Fig. 2 Overview of crystal structure of AceCas9–ΔHNH–sgRNA–DNA complex. a** Left: Primary structural features of AceCas9 and AceCas9–ΔHNH. BH, bridge helix (or arginine-rich helix), RuvC-I/RuvC-II/RuvC-III, RuvC domain, REC1/REC2, nucleic acid recognition domains 1 and 2, WED, wedge subdomain, TOPO, Topo-homology subdomain, PI, PAM-interacting subdomain. Right: DNA cleavage activity of AceCas9–ΔHNH with two single-guide RNA (sgRNA106 and sgRNA94). The cleavage experiment has been repeated more than three times with similar results. **b** Left: Secondary structure of the single-guide RNA, sgRNA106 and sgRNA94 (black), and their relationship with the DNA oligos (red) used in co-crystallization. Boxed nucleotides are present in sgRNA106 but not in sgRNA94. The nucleotide numbering is shown for both sgRNA106 and sgRNA94 except those in parentheses are for sgRNA94. Right: The omitted 2$F_o$–$F_c$ electron density contoured at 2.0σ around all nucleic acid components of the complex using the final refined ternary complex structure for phases and observed structural factors for amplitudes. The sgRNA94 is colored in black and the DNA is in red. **c** The structure of the AceCas9e–sgRNA94–DNA in a front and a back view. Domains and the nucleic acids are colored as in (**a**) and throughout the article. The DNA cleavage experiments shown in panels (**a**) have been repeated more than three times with similar results. Source data are provided as a Source data file.

nucleotides including the first triphosphate nucleotide (G1), and all of the DNA nucleotides (Fig. 2b, c). The sgRNA94 RNA forms the mFold-predicted secondary structure with its guide base-paired with the 20-nt complementary TS DNA. The nucleic acid complex is surrounded by the multi-domain AceCas9 protein. Similar to other Cas9, AceCas9 contains the conserved RuvC domain that is connected by the arginine-rich bridge helix (BH) to the nucleic acid recognition domain (REC) and the PAM-interacting domain (PID) (Fig. S3). The BH helix threads through the helical interior of the sgRNA and the well-conserved RuvC domain is poised to interact with the NTS DNA (Fig. 2c). Unlike the Type II-A Cas9s but similar to Type II-C Cas9s, AceCas9

contains two (REC1 and REC2) rather than three subdomains within its REC (REC1, REC2, and REC3) (Fig. 2c and Fig. S3). Its PID, although similarly divided into the WED, TOPO, and PI subdomains as in other Cas9s, contains unique insertions and wraps around the PAM helix differently (Fig. 2c and Fig. S4). Overall, AceCas9 resembles CdiCas9 (PDBid: 6JOO) in structure the most with a root mean square deviation (RMSD) of 2.6 Å for 861 Cα atoms (Fig. S3). AceCas9 also structurally resembles the ligand-free AnaCas9 (PDBid: 4OGE, RMSD: 5.4 Å for 861 Cα) and NmeCas9 (PDBid: 6JDV, RMSD: 7.1 Å for 438 Cα or PDBid: 6JE3, RMSD: 9.4 Å for 650 Cα) to a less degree (Fig. S3). However, the AnaCas9 structure is of the apo form and its higher

RMSD, therefore, mostly reflects the difference in domain rearrangement. The regions identified in the AceCas9 structure serve as excellent guides to protein engineering efforts in altering AceCas9 activity.

**Interaction with sgRNA.** The sgRNA94 RNA is comprised of a 20 nucleotide (nt) guide (G1-U20), the repeat:anti-repeat duplex (G21-U32 and A37-C48) connected by a GAAA tetraloop (G33-A36), a two nucleotide connecting loop (A49-A50) (CL1), stem loop 1 (G51-C67) (SL1), a second connecting loop (G68-C76) (CL2), and stem loop 2 (A77-U92) (SL2) (Fig. 2b). We analyzed its interactions with AceCas9 by its buried surface areas, close contacts, and a library-based functional assay in *E. coli* cells. The computed buried surface area and the close contacts of each nucleotide were obtained from the AceCas9–sgRNA–DNA structure coordinate. The library-based functional assay[31] was carried out by transforming a plasmid pool that co-expresses randomized sgRNA and the wild-type AceCas9 into *E. coli* cells harboring the *ccdb* toxic gene followed by sequencing the survival sgRNA (Fig. 3a).

The library-based functional assay indicated that AceCas9 requires a correct secondary structure of the sgRNA. The fact that

only ~9% of the unique sequences in the starting library were found in the survivor pool while both had nearly equal number of total sequences (Supplementary Data File 2) indicated that AceCas9 prefers the wild-type RNA sequence. The computed wild-type frequencies, or conservation scores, support this preference (Fig. 3a and Supplementary Data File 2). The few unconserved nucleotides are found scattered among the loops and the stems. For those unconserved within the stems, their base-paired nucleotides are also unconserved and vice versa, suggesting a functional requirement for the stems. The three loops, CL1, most of CL2, and SL2 contain highly conserved nucleotides (Fig. 3a). In comparison with the structure, some conserved regions are also highly buried. For instance, the first half of the repeat-anti-repeat stem is enclosed by the Bridge Helix (BH) and REC1. On this stem, the section comprised of G21-G24 and C44-G47 have high conservation scores. However, the reverse is not true. Nucleotides that are likely to survive are not necessarily buried by AceCas9. These regions include the base pairs toward the end of the three stem loops. For instance, U32-A37 base pair has a high conservation score but is not contacted by AceCas9 (Fig. 3b). Conservation of these regions, therefore, reflects the requirement for the sgRNA fold. Note that the highly conserved

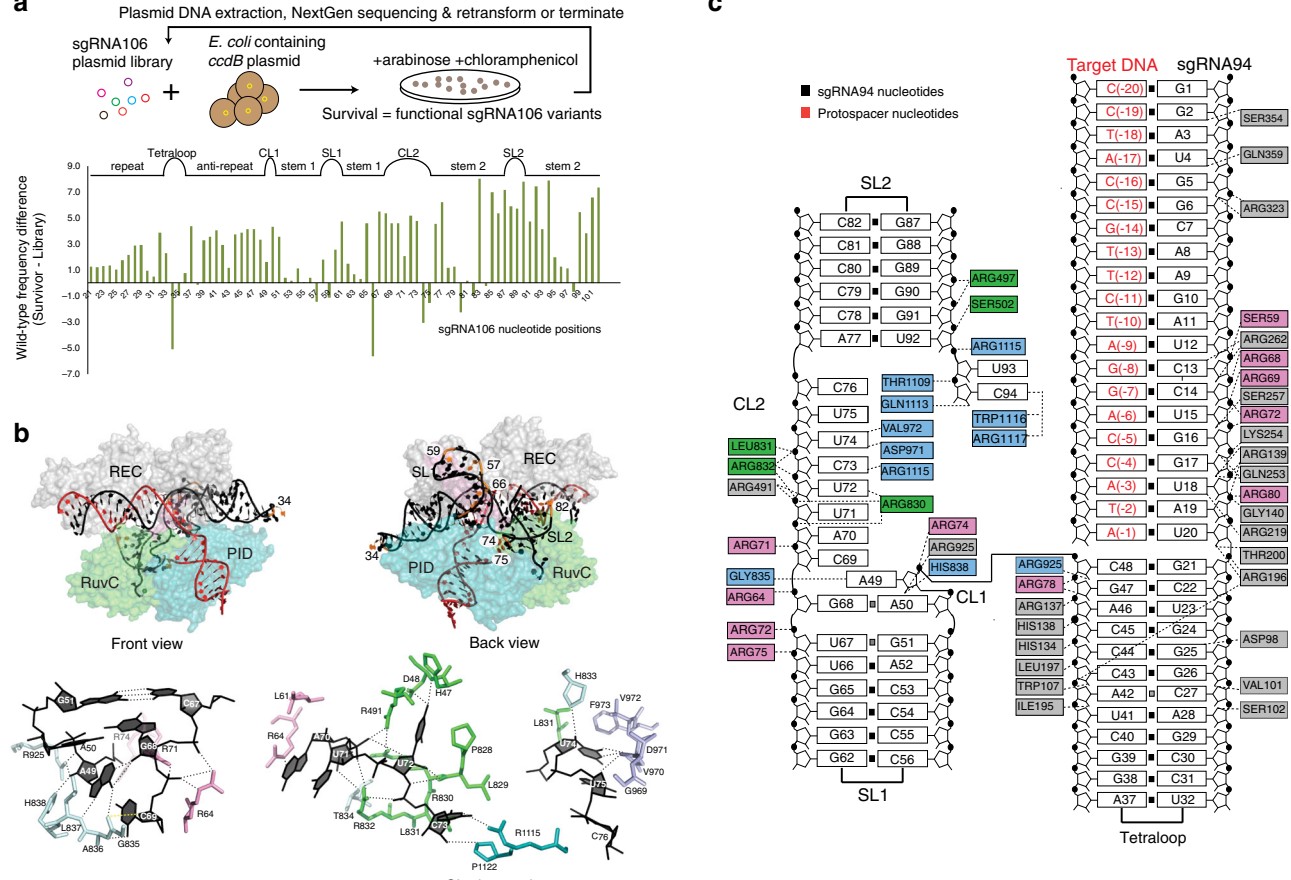

**Fig. 3 Recognition of sgRNA by AceCas9. a** Top: Schematic of the functional RNA selection procedure for identifying nucleotides critical to sgRNA function. Plasmids encoding both AceCas9 and a library of sgRNA106 were transformed into *E. coli* hosting a target plasmid that expresses the toxin ccdB. Functional variants of sgRNA were identified by sequencing the sgRNA library of the survival colonies. Bottom: Plot of the wild-type nucleotide read frequency difference between that of the survivor and that of the input libraries against the nucleotide positions. A low difference frequency indicates a high mutability and vice versa. Secondary structure elements are shown on top and denoted in same terms as in Fig. 2. **b** Top: Selected positions of highly varied nucleotides (orange) on the 3D structure in both front and back views. Bottom: Detailed protein–RNA interactions between connecting loops, CL1 or CL2, and AceCas9. RNA nucleotides are colored in black. Protein residues that within 3.6 Å from the displayed RNA nucleotides are displayed as sticks and are colored as in Fig. 2. Dashed lines denote polar interactions that are within 3.4 Å. **c** Detailed close-contact interactions between sgRNA and AceCas9–ΔHNH. Amino acids are colored according to the colors used for the domains in Fig. 2.

region in SL2 identified by the library-based assay was not included in crystallization. Should it be present, it could interact with AceCas9, possibly with REC2. Otherwise, its conservation again reflects the requirement for the specific secondary structure of the guide RNA by AceCas9.

Base-specific contacts between sgRNA and AceCas9 are surprisingly few and are found almost exclusively within the two connecting loops. The CL1 and CL2 nucleotides do not form Watson–Crick base pairs and are instead nestled in the interior of AceCas9 with high buried surface areas (Fig. 3b and Supplementary Data File 2). Within CL1, the N6 atom of A49 is contacted by carbonyl oxygen of Gly835 in the linker between the RuvC and the PID domains. The downstream A50 forms a sheared A–G pair with G68. Consistently, A49, A50, and G68 showed high conservation scores from the library screening (Fig. 3a and Supplementary Data File 2). For CL2, Asp48, and Arg830 of RuvC, Arg1115, Asp971, and Val972 of the PI subdomain make specific contacts with the bases of U72, C73, and U74, respectively (Fig. 3b). Consistently, the library screening also identified U72, and C73 to be invariant and U74 to be exclusively U or C for function (Fig. 3a and Supplementary Data File 2). Thus, the bases within CL1 and CL2 are important for AceCas9 assembly and function. AceCas9 also makes extensive but non-specific contacts with the guide region. Although several base-specific contacts by AceCas9 are observed within the guide region (G2, C13, and U18), these are with the minor groove edge of the bases and thus do not restrict the sequence of this region (Fig. 3b), making AceCas9 adoptable for targeting any protospacers.

**Recognition of the 5′-NNNCC-3′ PAM.** The ternary complex crystal structure reveals specific interaction of the 5′-NNNCC-3′ PAM with AceCas9 residues in its PI subdomain (Fig. 4). Specifically, the carboxylate group of Glu1044 contacts the exocyclic amino of C4* on NTS and the guanidinium groups of Arg1088 and Arg1091 contact the exocyclic oxygen of G(-4) and G(-5) on TS, respectively. In addition, Arg1091 establishes a salt bridge with Glu1044 to further enhance the Glu1044-C4*-Arg1091-G(-5) network of interactions (Fig. 4a). This PAM-interaction network explains the specificity of AceCas9 for the 5′-NNNCC-3′ PAM and its methylation state, although it does not explain why 5′-NNNAC-3′ is a weak PAM. Mutation of Glu1044 or Arg1091 significantly reduced the DNA cleavage activity while that of Arg1088 moderately reduced the activity (Fig. 4b, c). Note that C5* on NTS does not establish any close contact with AceCas9 residues despite the deleterious effect on cleavage when it is mutated to other nucleotide[29]. This can now be explained by the observed interactions between AceCas9 and G(-5) that pairs with C5*.

The currently eight available Cas9–sgRNA–DNA complex structures show a conserved fold of the PI subdomain but a divergent method of PAM recognition (Fig. S4). The PI subdomain is characterized by a central β-sheet of six anti-parallel β-strands (β4–β9) with widely varied insertion helices. In most cases, PAM-interacting residues are located within β5, β6, and β7 while in one case (FnoCas9), a PAM-interacting reside is in the insertion helix, and in another (Nme1Cas9), a PAM-interacting residue is found in the neighboring TOPO subdomain (Fig. S4). In order to more systematically assessing the importance of the structural elements involved in AceCas9 function, we subjected the PI subdomain of AceCas9 (residues 1024–1107) to a protein functional selection assay similar to that for sgRNA (Fig. 3a). The estimated $10^5$ AceCas9 PI variant clones were introduced to the *ccdb*-expressing cells and the surviving clones (active AceCas9 variants) were harvested for next-generation sequencing analysis. We first computed the difference frequency for each of the 20 amino acids excluding the wild-type amino acids to appear at each position between the input and the survivor pools (Supplementary Data File 3), where a small and a large value indicate favored and disfavored non-wild-type amino acid. We then plotted the difference frequency of the wild-type amino acid between the survivor and the input pools where a high value indicates conservation of the wild-type amino acid (Fig. 4d). Finally, we listed frequencies of all 20 amino acids in both the input and survivor pools at four selected positions (Table S2). The region between β5 and iα3 seems to have an overall high difference frequency and is thus critical to PAM recognition (Fig. 4d). Furthermore, the three residues that directly contact PAM, especially Glu1044, showed a strong preference for the wild-type amino acids (Fig. 4d, Table S2, and Supplementary Data File 3). In contrast, a number of residues (D1025, S1027, F1029, and P1034) on the first insertion helix, iα1, which resides under the PAM duplex, show a preference for non-wild-type amino acids (Fig. 4d and Supplementary Data File 3). We also found that Leu1075 on the loop of connecting insertion helices iα2 and iα3 has a strong preference for valine (Fig. 4d, Table S2, and Supplementary Data File 3). These regions are, thus, more prone to mutations.

To further analyze the roles of the three PAM-interacting residues, Glu1044, Arg1088, and Arg1091, on recognition of other PAM sequences, we performed in vitro DNA cleavage activity assays on a series of PAM variant substrates with the wild-type and AceCas9 mutants (Fig. 5 and Fig. S5). The PAM variants included 5′-NNNAC-3′, 5′-NNNGC-3′, 5′-NNNCG-3′, 5′-NNNTC-3′, 5′-NNNCT-3′, and a number of variants within the first three nucleotides (5′-NNN). In addition, both the supercoil and pre-linearized forms of the PAM variants were tested to assess the level of activities. The wild-type enzyme has the strongest activity on supercoil DNA with either the 5′-NNNAC-3′ or 5′-NNNCC-3′ PAM while weaker activity on their pre-linearized forms (Fig. 5 and Fig. S5). R1088A seems to retain a significant portion of the wild-type activities on both DNA forms while E1044A has some activities on supercoiled DNA. In contrast, R1099K or R1088A/R1091A double mutant has nearly no dsDNA cleavage activity but some nickase activities (Fig. 5 and Fig. S5). These results are consistent with the observed close contacts involving the three residues. We note a slightly different pattern of dependence on PAM sequences between the wild-type and R1088A especially on pre-linearized DNA (Fig. 5 and Fig. S5), suggesting a possibility to alter the PAM specificity of AceCas9 through engineering Arg1088. We further tested the four mutants on DNA oligo targets with the $^{5m}$C4*-containing PAM and found negligible activities similarly as with the wild type (Fig. S6), suggesting that the $^{5m}$C4*-containing PAM is also deleterious to PAM recognition by the mutants.

Finally, to directly observe the impact of PAM sequence on AceCas9 interactions, we co-crystalized AceCas9, sgRNA94, and a DNA target containing the 5′-NNNTC-3′ PAM, or C4*T mutant, under a nearly identical condition as that of the wild-type complex. The C4*T (TC PAM) crystal diffracted to 3.61 Å and has a shorter b-axis than that of the wild-type complex crystal (112 versus 119 Å), indicating a structural change in the complex. While a non-PAM sequence in a dsDNA likely prevents unwinding and subsequent formation of the R-loop, the artificially constructed C4*T oligos lacking the paired protospacer helix in our co-crystallization studies are attached to AceCas9 RNP largely via the guide-TS heteroduplex, which allowed us to examine how the TC PAM may interact with AceCas9 PID. Superimposing of the refined C4*T to that of the wild-type complex structure revealed rotations of both the PAM DNA helix and the REC2 domain toward each other (Fig. 6). As a result, the

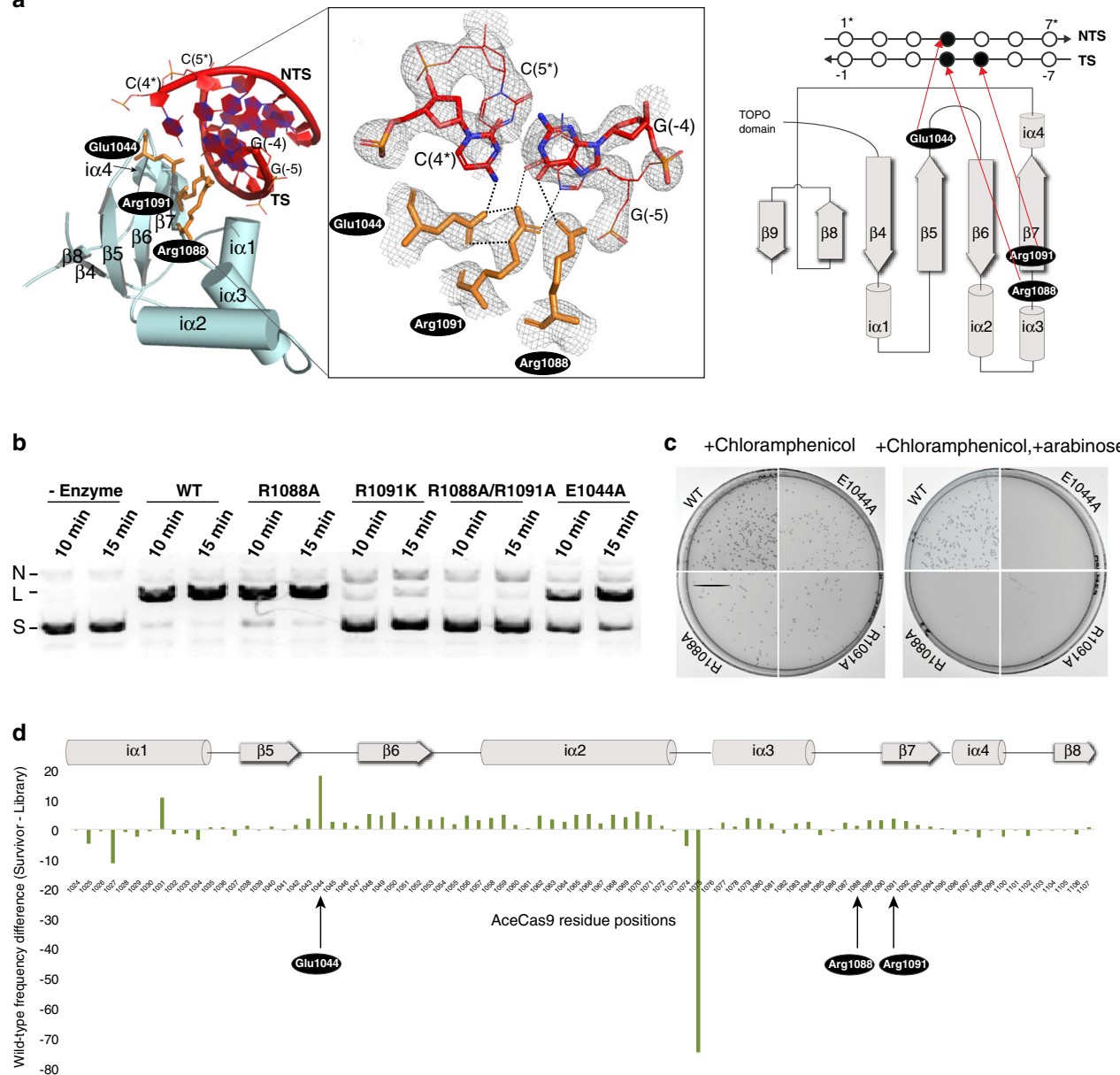

**Fig. 4 Recognition of the protospacer-associated motif (PAM) by AceCas9. a** Left: The structure of the PAM-interaction (PI) subdomain and its detailed interactions with the 5′-NNNCC-3′ PAM. Inset: close-up view of PI-PAM interactions overlaid with the omit $2F_o–F_c$ map of the region. Right: Topology of the AceCas9 PI subdomain, location of the key PAM-interaction residues and the schematic interactions with the PAM nucleotides. Secondary structure elements are labeled where "iα" denotes insertion helices. **b** In vitro DNA cleavage results with the wild type (WT) and the mutants of the three PAM-interacting residues. "L" denotes linearized product DNA, "S" denotes the supercoil DNA substrate, and "N" denotes nicked DNA. Two reaction times for each enzyme used are indicated. **c** Cell survival assay results of the wild-type (WT) and three PAM-interaction mutants. **d** Plot of the wild-type amino acid frequency difference between that of the survivor and that of the input libraries against the residue positions. A low difference frequency indicates a high mutability and vice versa. Secondary structure elements are shown on top and denoted in same terms as in panel (**a**). The DNA cleavage experiments shown in panels (**b**) and the survival assay shown in (**c**) have been repeated more than three times with similar results. Source data are provided as a Source data file.

three residues that contact the PAM in the wild-type complex shifted up to 5.5 Å away from the 5′-NNNTC-3′ PAM (Fig. 6). We infer that should AceCas9 encounters a dsDNA with the TC PAM, it would have a weak interaction with TC, thus preventing local unwinding and cleavage of the dsDNA substrate.

## Discussion

We characterized AceCas9–ΔHNH–sgRNA–DNA complex structure at a 2.9 Å resolution and identified the molecular basis

for its high specificity for the 5′-NNNCC-3′ PAM. AceCas9 residues directly contact C4* of the nontarget strand and G(-5)-G(-4) of the target strand. Accordingly, methylation of C4* but not C5* abolishes the DNA cleavage activity of AceCas9. To our knowledge, AceCas9 is uniquely sensitive to DNA methylation in its PAM. On the other hand, the widely used SpyCas9 is not sensitive to DNA methylation[16,18]. AceCas9 may thus find unique applications where sensitivity to DNA methylation is required.

DNA base modification is a universal phenomenon in life and expands the relatively simple alphabet (A, T, C, and G) with an

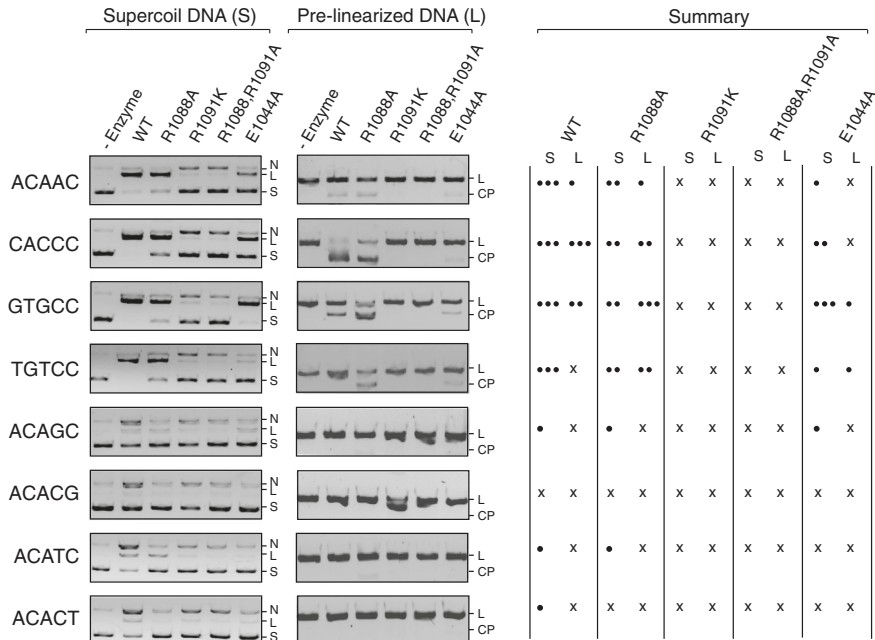

**Fig. 5 Protospacer-associated motif (PAM) recognition by AceCas9 mutants.** In vitro cleavage of the supercoil and pre-linearized plasmid substrates with different PAMs by the wild type and mutant AceCas9. "L" denotes linearized product DNA, "S" denotes the supercoil DNA substrate, "N" denotes nicked DNA, and "CP" denotes cleaved product. Each DNA substrate at 6 nM concentration was incubated with 1 μM AceCas9 or its mutants at 50 °C for 15 min followed by gel separation and ethidium bromide staining. The intensity of each reaction product was estimated and summarized in the summary table where the number of filled circles indicates the activity level and "x" indicates non-detectable activity. The DNA cleavage experiments were repeated more than three times with similar results. Original images of this figure are provided in Supplementary Fig. S5 with DNA markers.

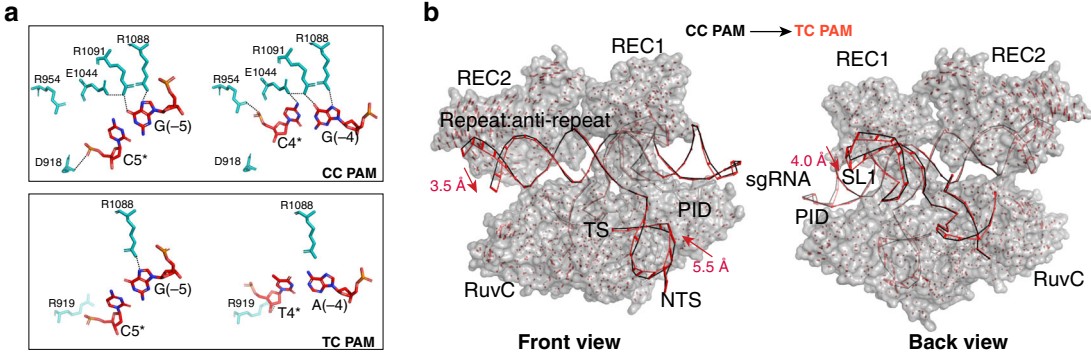

**Fig. 6 Crystal structure of AceCas9–ΔHNH–sgRNA–DNA (C4*T) complex. a** Detailed comparison of PI-PAM interactions between the 5′-NNNCC-3′ (CC PAM) and the 5′-NNNTC-3′ (TC PAM) complex. AceCas9 residues within 3.6 Å of the four PAM nucleotides are shown in teal color. Dashed lines indicate the close contacts among the protein residues and between the protein and the PAM nucleotides. Note the significantly fewer number of residues in the TC PAM complex (bottom) than that in the CC PAM complex (top). **b** Comparison of the overall structures between the 5′-NNNCC-3′ (CC PAM) and the 5′-NNNTC-3′ (TC PAM) complex. Pair-wise difference vectors were computed for the carbonyl carbon (for proteins) and the phosphate atoms (for RNAs) between the two structures and displayed as red bars on the wild-type structure in surface (protein) and ribbon (nucleic acids) representations. Nucleic acids are shown in both complexes and are colored black (CC PAM) and red (TC PAM), respectively. Labels of domains and RNA secondary elements are defined as in Fig. 2.

additional layer of complexity. Bacteria use DNA modification to protect their genome from being removed by their own immune processes or to regulate gene expression[32,33]. DNA modification also plays an important role in epigenetic control in eukaryotes[34]. For instance, 5mCpG methylation has been shown to be bio-markers for cancers[35] while non-5mCpG methylation can regulate brain functions[36,37]. Although the specific sensitivity of AceCas9 to 5mCpC does not align with the most abundant 5mCpG methylation in normal mammalian cells[38], non-5mCpG methy-lation is significant in plant cells[39] as well as stem cells[37,40]. In these cases, while the sensitivity of AceCas9 to 5mCpC should be taken into consideration in gene editing, it may be explored for

detection of epigenetic changes. It is also possible, through pro-tein engineering, to alter AceCas9 sensitivity for other methylated sequences including 5mCpG.

It is remarkable that, despite a low sequence identity in both the protein and the sgRNA, currently known Cas9s show many conserved structural features. They have the same four-domain protein architecture and similar sgRNA secondary structures. However, specific elements encoded within the common fold and secondary structures contribute to the unique functional prop-erties of each Cas9. Unlike other Cas9, AceCas9 functions at an elevated temperature, interacts with its specific RNA partner, and recognizes the unique 5′-NNNCC-3′ PAM. Our work provides

structural data that explains how AceCas9 interacts specifically with its RNA and PAM. However, it does not provide sufficient insights into its thermostability, neither from the spatial distribution of charged amino acids nor disulfide bonds because these attributes of AceCas9 do not deviate significantly from those of the mesophilic Cas9 (Fig. S3). Other biophysical studies on the dynamics of AceCas9 may shed light on the origin of its thermostability.

To probe more comprehensively the protein and sgRNA elements required for these functions, we applied a previously established functional selection method to study AceCas9–sgRNA and AceCas9–PAM interactions. The library-selected functional sgRNA variants show a requirement for the two stems (SL1 and SL2) as well as the repeat:anti-repeat stem with only a few variable positions in loops. However, many unpaired nucleotides are just as important as the paired regions. The observed pattern of RNA conservation, especially in the unpaired regions, matches that identified from analyzing the close contacts between the protein and sgRNA in the ternary complex, suggesting that AceCas9 has evolved to recognize both the secondary structure and the bases of its guide RNA.

Despite low sequence identity, all known Cas9s share a conserved PI subdomain comprised of a central six-strand antiparallel β-sheet (β4–β9) interspaced with a-helical insertions. Most PAM-interacting residues are scattered within the segments of β5, β6, and β7. It is the amino acid identify and their specific location within the three β-strands that determine the PAM specificity. Our functional selection experiment confirms the importance of the identity of the three PAM-interacting residues. Furthermore, we identified that most critical region of PI subdomain extends from the loop connecting β5 and β6 to the end of the insertion helix iα2.

We previously engineered AceCas9 variants that have different specificity and catalytic efficiency than the wild type[41]. The refined AceCas9–ΔHNH–sgRNA–DNA structure provides a satisfying model to explain our engineered variants. We validated the interactions involved with the phosphate lock residues that were found to be critical to specificity in the seed region[41]. As we predicted, Glu839 and Glu840 interact with the sharply bent phosphate backbone in the TS DNA and their substitutions by arginine and tyrosine, respectively, compromises the fidelity of AceCas9 in recognizing the first RNA–DNA base pair, likely due to the increased buried surface area of the TS DNA[41].

## Methods

**Sample preparation and crystallization.** The DNA sequence encoding the full-length AceCas9 protein was codon-optimized for bacterial expression and cloned between Nco1 and BamH1 restriction sites in a pET28 expression vector (GenScript, Piscataway, New Jersey). A truncated version whose HNH catalytic domain (residues 523–679) is substituted with a short linker (523GGGSAGG529) (AceCas9–ΔHNH) was created using the Q5-mutagenesis kit (New England Biolabs, Ipswich, MA) (Fig. 2a) with primers listed in Table S3. Other AceCas9 mutations were also carried out using the Q5-mutagenesis kit (New England Biolabs, Ipswich, MA) with primers listed in Table S3. After expression in BL21 cells, AceCas9 and its variants were purified via nickel-affinity and Heparin ion-exchange chromatography. The full-length protein used for activity assays was further purified by size-exclusion chromatography and concentrated to 10 μM for storage at −80 °C. The AceCas9–ΔHNH fractions from Heparin column were pooled and stored at −80 °C before crystallization. The single-guide RNA (sgRNA106 and sgRNA94) used for crystallization were produced by in vitro transcription with the T7 RNA polymerase (gift from A. Pyle) and purified on an 8% denaturing urea polyacrylamide gel electrophoresis (PAGE) gel (Fig. 2b). Both the target and the nontarget DNA strands were purchased from Eurofins Genomics (Louisville, KY). Crystallization complexes were formed by incubating the Heparin-column purified AceCas9–ΔHNH with sgRNA94, target and nontarget DNA strands in a 1:2:2.8:3.4 ratio at 50 °C for one hour before being purified on a Superdex 200 increase column (GE Healthcare). The ternary complex was then concentrated until absorption at 260 nm (A260) reached ~60 arbitrary units and used for crystallization. The seleno-methionine labeled protein was expressed in *E. coli* B834

(DE3) methionine auxotroph cells and purified and assembled similarly as the native AceCas9–ΔHNH protein.

The initial AceCas9–ΔHNH–sgRNA94–DNA complex was crystallized at 30 °C by sitting-drop vapor diffusion technique as implemented in Crystal Gryphon (Art Robbins Instrument, Sunnyvale, CA), and was optimized manually by the hanging-drop method. Both the unlabeled and the seleno-methionine-labeled crystals were obtained by mixing the complex in a 1:1 ratio with a reservoir solution containing 0.04 M Citric acid, 0.06 M Bis-tris propane, and 15–20% polyethylene glycol 3350. Crystals grew to a maximal size at 30 °C in 3–5 days. Freezing of crystals was optimized by a series of crystal-annealing steps with cryo-protecting solutions containing the mother liquor plus 5%, 15%, and 30% glycerol, respectively, in 3–15 min. Most of the diffraction data were collected from the frozen crystals at the 24-ID-C and 24-ID-E beamlines of The Northeastern Collaborative Access Team (NECAT) at the Advanced Photon Source and others were collected at the 17-ID-1 or 17-ID-2 beamlines at the National Synchrotron Light Source II (NSLS-II) synchrotron. For each of the heavy metal-labeled crystals, whether used or not in final phase determination, X-ray wavelength was tuned to its anomalous edge and a full sweep of 600–720° rotation images were collected continuously in a shutterless mode. At NECAT, the RAPD molecular package of programs was used for automated data processing with XDS[42,43]. At NSLS-II, datasets were integrated and scaled in real time by a modified FAST_DP[42,44–46]. For all data collected, reflections with <I/s(I)> as low as 0.9 and an overall CC1/2 of 99.7% and better were kept because they noticeably improved electron density and refinement statistics. The final data collection statistics is provided in Table S1.

**Structure determination and refinement.** The phases were determined by a combination of single-wavelength anomalous dispersion and molecular replacement (SAD-MR) using the program suite Phenix[47]. A model of AceCas9–ΔHNH protein was first made with the MRage program module[48], which, along with an independent model of the sgRNA based on the CdiCas9 structure (PDBID: 6JOO) and an ideal DNA duplex representing the 10-base-paired PAM region, were used as multiple ensembles in a molecular replacement search with PHASER[49]. A single solution with a high log-likelihood gain (LLG) score (1850) with the MRage model of AceCas9–ΔHNH, that of sgRNA:complementary DNA, and that of the PAM region was obtained that reconstructed the typical ternary complex architecture of Cas9, suggesting the validity of this solution. The MR solution was subsequently used to assist heavy-atom site search in the Se-SAD data set, which led to a convincing solution with all six selenium sites identified from the anomalous difference Patterson map. Phases were further improved by solvent flattening and then used to compute an electron density with the observed structural factors, which showed sufficient quality to allow tracing of the protein and nucleic acid residues. All 6 selenium peaks matched the predicted positions for the methionine residues. Initial refinement was carried out against the Se-SAD data set by using the phased maximum likelihood (MLHL) target, which improved the structure significantly. The final structure was refined against a native data set at 2.9 Å resolution to a satisfactory residual factors and stereochemistry values (Table S1). Model building was carried out with the program COOT[50]. Figures were prepared with the program PYMOL[51].

**In vitro cleavage assay.** In vitro cleavage assay was carried out as previously described[29]. Briefly, AceCas9 RNP (1 μM) was incubated with target DNA (~6 nM) for 60 min at 50 °C in a cleavage buffer (20 mM Tris pH 7.5, 150 mM KCl, 2 mM DTT, 10 mM MgCl₂, 5% glycerol). The reactions were stopped by adding the running buffer (25 mM Tris pH 7.5, 250 mM EDTA pH 8.0, 1% SDS, 0.05% w/v bromophenol blue, 30% glycerol), resolved on a 0.5% agarose gel containing ethidium bromide, and imaged using Chemidoc XRS System (Bio-Rad, Hercules, CA). The pre-linearized plasmid substrates were obtained from restriction enzyme digestion, heat inactivation of the enzyme, and subsequent AceCas9 digestion. The plasmid DNA used for 5mC4* or 5mC5* methylation contain GGC4*C5* or C4*C5*AGG PAM sequence that is the target for the *HaeIII* or the *dcm1* methyltransferase. Since *HaeIII* is not present in *E. coli* while *dcm1* is, 5mC4*-containing plasmid was obtained by *HaeIII* treatment following manufacturer's instruction and 5mC5* would be present following amplification in *E. coli*. The presence of methylation on each cytosine was confirmed by *HaeIII* restriction treatment (Fig. 1d) or bisulfite sequencing (Fig. S1). The oligo DNA used for 5mC4* or 5mC5* methylation was purchased from Eurofins Genomics (Louisville, KY), annealed with excess the non-modified nontarget strand before being used in AceCas9 cleavage reaction.

For binding competition experiments, increasing concentration (12.5–2500 nM) of competitor DNA duplex (with no, 5mC4*, or 5mC5* methylation) were added to the typical plasmid cleavage experiment. The fraction of cleavage was fitted to a competitive inhibition model in Prism 8.1.1 to extract $K_I$.

**Cell survival assay.** The cell survival assay for AceCas9 was performed as previously described[29]. Briefly, electro-competent BW25141 *E. coli* (gift from D. Edgell) containing a *ccdB* plasmid were transformed with 100–200 ng of the AceCas9 or its variant-encoding plasmid and were recovered in the super-optimal broth media (SOC) for 30 min at 37 °C with shaking. Protein expression was induced with 0.05 mM Isopropyl β-D-1-thiogalactopyranoside (IPTG), followed by

an additional recovery at 37 °C for 1 h with shaking. The cells were then plated in chloramphenicol (52 μg/ml) or chloramphenicol-arabinose (10 mM) containing plates, and were incubated for ~16 h at 37 °C. Plates were then imaged and survival percentages were calculated as the quotient of the number of colonies on the chloramphenicol-arabinose plate and that on the chloramphenicol-only plate.

**Plasmid depletion assay and data analysis**. A pUC19 plasmid bearing the target spacer sequence followed by seven randomized base pairs was subjected to cleavage by AceCas9. As a control, the plasmid library was incubated with AceCas9 storage buffer. The uncleaved populations from both the AceCas9- and the buffer-treated plasmids (supercoil) were gel extracted (Qiagen) and used for amplicon NGS library creation (see below). The libraries were subjected to 2 × 150 cycle paired-end sequencing using MiSEQ (Illumina). Both 5′ and 3′ ends of raw data output from the sequencer were trimmed with an error probability limit of 0.05 and aligned to reference sequence using a medium-low sensitivity, iterative fine-tuning up to five times, and a minimum mapping quality score of 30. A Python program modified from that by Kleinstiver et al.[52] was created to analyze the sequence read frequencies in sets of 2–7 nucleotides in both the AceCas9- and buffer-treated libraries. For each length segment, a given sequence combination was counted and normalized by diving the total counts of the same segment to obtain its frequency. A PAM depletion score was then obtained by calculating the ratio of cleaved to uncleaved frequency. A low score indicates efficient depletion by the AceCas9 treatment, allowing identification of the PAM. The depletion scores for all dinucleotides at positions 4* and 5* were plotted in Fig. 1a that identified 5′-NNNCC-3′ is the most favorable PAM.

**Plasmid library preparation**. Libraries used for directed protein or RNA evolution were prepared as previously described[31]. Briefly, targeted locations were amplified with Gibson Assembly primers (Table S3) via error-prone PCR (to produce an average of 1.5 mutation per 100 bp) and inserted back into the pACYCDuet-Ac9-g106 vector[29]. Assembled plasmids were transformed into DH5a cells, and ~60,000 colonies were pooled the next day into chloramphenicol containing Luria-Bertanni media. Following 2 h of incubation, DNA was extracted via maxi-prep (Qiagen) and concentrated to ~200 ng/μL.

**Next-generation sequencing and data analysis**. The randomized region of each survival pool was PCR amplified with primers containing Illumina indexes. After addition of adaptors, pooled libraries were subjected to 300 cycle single-end sequencing using MiSEQ (Illumina, San Diego, CA). Sequences were aligned and analyzed using a suite of in-house programs. In both the RNA and PID functional assays, the script listed all sequences in both the input and survivor pools and kept those containing the wild-type length and at least one mutation in the randomized region. The sequences with no mutation were counted but not used in subsequent analyses. For the RNA library, the frequency for each of the four nucleotides to appear in either the input or the survivor pools was computed and compared. For the PID library, the nucleotide sequences were first translated to amino acids before analysis. Among all tabulated non-wild-type sequences, those with a single amino acid change comprise ~68% and those with two mutations comprise another ~20%. The frequency of the wild-type nucleotide or amino acid at each position was computed and the difference between that of the last survivor pool library and that of the input library was obtained.

## Data availability
The atomic coordinates have deposited at Protein Data Bank with accession codes 6WBR for the wild type and 6WC0 for the C4T PAM complex, respectively. The data associated with the next-generation sequencing analysis is given in the Supplementary Datasets 1–3 and Supplementary Table 2. Any additional raw data associated with all the assays can be provided upon request. Source data are provided with this paper.

## Code availability
The suite of Python programs used for library sequencing data analysis is available upon request.

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

## Acknowledgements

We thank B. Washburn, C. Pye, and Kristina Poduch of the FSU Molecular Cloning Facility for cloning experiments, J. Sefcikova in assisting RNA purification, S. Thayu-manasamy for assistance in data collection and Miller and A. Brown of the FSU Sequencing facility for assistance with NGS library preparation and sequencing, D. Edgell for supplying BW25141 cells, A. Pyle for supplying T7 RNA polymerase clone. This work was supported by NIH grant R01 GM099604 to H.L. This work is based upon research conducted in part at the Northeastern Collaborative Access Team beamlines, which are funded by the National Institute of General Medical Sciences from the National Institutes of Health (P30 GM124165). The Pilatus 6M detector on 24-ID-C beam line is funded by a NIH-ORIP HEI grant (S10 RR029205). The Eiger 16M detector on 24-ID-E beam line is funded by a NIH-ORIP HEI grant (S10OD021527). This research used resources of the Advanced Photon Source, a U.S. Department of Energy (DOE) Office of Science User Facility operated for the DOE Office of Science by Argonne National Laboratory under Contract No. DE-AC02-06CH11357. This research also used resources of the National Synchrotron Light Source II, a U.S. Department of Energy (DOE) Office of Science User Facility operated for the DOE Office of Science by Brookhaven National Laboratory under Contract No. DE-SC0012704. The Life Science Biomedical Technology Research resource is primarily supported by the National Institute of Health, National Institute of General Medical Sciences (NIGMS) through a Biomedical Technology Research Resource P41 grant (P41GM111244), and by the DOE Office of Biological and Environmental Research (KP1605010).

## Author contributions

A.D. and H.L. designed all experiments, A.D. purified AceCas9, crystallized the complex, and collected data, H.L. solved the structure, T.H. designed primers for AceCas9 plasmid libraries and prepared DNA for NGS experiments, C.S. performed in vitro DNA cleavage assays, E.W. assisted in methylation cleavage experiments, M.Z. analyzed NSG data, A.D. and H.L. analyzed structural and biochemical data, wrote, and edited manuscript.

## Competing interests

The authors declare no competing interests.
