## [Peer Review File · Nature Communications]

REVIEWER COMMENTS

Reviewer #1 (Remarks to the Author):

Das et al., describe the recognition of an NNNCC PAM by AceCas9. Using biochemistry, the authors demonstrate that AceCas9 is sensitive to the methylation status at the C4 position within the PAM. The authors then describe the overall structure of AceCas9, the molecular details around its sgRNA recognition and the molecular basis for discriminating methylation at the C4 position. The biochemistry and structural biology are clear, and the conclusions fitting given the data, however, the manuscript would benefit greatly from improved clarity in sections and elaboration on important points as described below. Overall, the manuscript is thorough, and the data reasonably well presented throughout to give a narrative that would be of broad interest to the community provided the authors substantiate some claims.

Major Comments

1] A major focus of the manuscript is the described molecular details of the AceCas9 interaction with the NNNCCNN PAM (Figure 4). The authors adequately describe the interactions in the main text and Figure 4A and 4B are very clear. However, given the importance of the observed interactions to the narrative, it is critical that the authors include either a main text or supplementary figure panel demonstrating the quality of the electron density that supported modelling in this region. Specifically, the authors should use an unbiased map suggest as an omit map that excludes the region of interest described in Figure 4A. Depending on the clarity, the authors could include this overlaid in the boxed-out region of 4A or as an additional panel in a supplementary figure.

2] Figures 1 – 4 are well presented and described in the main text. However, the manuscript would benefit substantially from improvements to both the clarity of the text and the figures surrounding the data in Figure 5. Specifically, Figure 5 is referenced in a single sentence within the main text. However, there is a lot of data presented and it needs to be clearly described. Additionally, the gels appear to be spliced together and formatted differently which is concerning. If supported by the journal, the authors are encouraged to include uncropped images of all gels described in the manuscript for transparency.

3] It is the opinion of this reviewer that the TC-PAM containing structure is incorrectly described as an inactive state and would be more correctly described as an artefactually active state. From a purely biophysical perspective, the sole purpose of Cas9 recognition of a PAM is to provide a nuanced distortion to the otherwise B-form helical DNA that allows for strand invasion by the pre-ordered guide RNA spacer resulting in a stable R-loop structure. When the PAM is not consensus, strand invasion is not favored regardless of the spacer sequence and thus Cas9 does not form an R-loop and cut the dsDNA. To capture the TC-PAM structure, the authors used a DNA substrate that comprises a target strand containing the target nucleotides complementary to the guide RNA spacer in addition to the TC-PAM proximal nucleotides. In contrast, the non-target strand is short and only comprises the TC-PAM and its proximal sequence. In effect, the authors have provided an already unwound target strand that is completely accessible to the sgRNA spacer bound to AceCas9. It is therefore unsurprising that the AceCas9 can bind to the target strand and form a stable structure regardless of the PAM sequence. However, as the authors noted, the PAM proximal duplex is not tightly associated with the PI domain due to the lack of the consensus 'C' at position 4 of the PAM which is itself interesting to observe. However, the implications of the substrates used, and the observed lack of PAM interaction, may be lost on some readers. The authors would benefit from describing that the mutated PAM and lack of interaction would likely prevent R-loop formation in the first place and that the only reason they observe this state is because of the substrates used. Finally, the Figure 6A would benefit from some other features to better illustrate the distances between protein and nucleic acid. At present, it is not immediately obvious that the DNA in the TC-PAM structure is that much further from the AceCas9 PI domain.

4] The authors comment on how the sensitivity of AceCas9 to methylation at the C4 position presents opportunities for technology development for detecting epigenetic changes. However, as the authors describe, 70-80% of methylated cytosines occur at CpG sites. Based on the authors data, AceCas9 would not be able to discriminate this particular methylation site by PAM recognition since it does not match the consensus. The authors should comment on this and provide some elaboration on the potential utility of the PAM sensitivity to methylation marks in the discussion especially given how much emphasis is placed on this activity.

5] The gap between R_{work} and R_{free} is quite large (~5%). Normally this would suggest some amount of model bias if the phases were estimated by molecular replacement alone, however, the authors used both molecular replacement and experimentally determined phase information from SeMet-crystals. Could the authors comment on the large gap? Given the methods used, it suggests some degree of overfitting has occurred. Furthermore, the authors indicate on Page 8 in the results section that the Rfactors are 0.24/0.27 (R/R_{free}), which does not match the Table or the provided deposition files.

Minor Comments

Page 5 – The sentence “Here we further characterized the PAM recognized by AceCas9 and report crystal structures of a functional AceCas9 variant bound with its guide RNA and a DNA target” is somewhat misleading. The function of Cas9 is to bind and cut dsDNA. The variants crystallized are lacking the HNH domain and are therefore non-functional. This should be re-worded to clearly describe what is meant by ‘functional’.

Page 5 – The sentence “Mutation of the 5'-NNNCC-3' to 5'-NNNTC-3' severely impaired AceCas9 activity and caused detachment of AceCas9 from the PAM” is somewhat misleading. Detachment implies the active release of something that was once bound to AceCas9 after the introduction of the C-T change. Technically, the C-T change likely prevented stable binding in the first place rather than causing detachment. This should be reworded to better reflect the experiments and observations in the structure.

Page 5 – The sentence “The cleavage products by a functional AceCas9 or that lacking the guide RNA of a plasmid library containing a cognate protospacer followed by seven randomized base pairs were sequenced and analysed (Figure 1A)” is confusing. Consider rewording to make it clearer.

Page 6 – The sentence “We constructed eight plasmid DNA substrates containing various PAM sequences and subjected them to cleavage by AceCa9 both in their supercoil and pre-linearized forms” would benefit from the addition of ‘in vitro’ to make it clear that the assay uses purified proteins and nucleic acid substrates.

Page 9 – The sentence “The regions identified in the AceCas9 structure serve as excellent guides to protein engineering efforts in improving AceCas9 activity” seems out of place. Could the authors elaborate on what this means?

Page 13 – The authors describe a “protein directed evolution assay” however based on the data it is more accurate to describe this as a selection since there is no real ‘evolution’ happening here, just a selection for functional variants. The authors should describe it as something to the effect of a ‘functional AceCas9 selection where the input library contained 105 AceCas9 PI variants.’

Page 13 – The description of the data from Table S4 where the authors state that certain positions have preferences is confusing. Do the authors mean to say that the residues at these positions are tolerated in the described functional assay? Please consider rewording this section to improve clarity.

Page 16 – The sentence starting “Despite low sequence homology...”. Proteins being homologous is binary. They are either homologous or they are not. The correct term to use in this sentence would be low sequence “identity”. Sequence identity is used to define if something is homologous.

Page 19 – Sentence “For each of the heavy metal-labeled protein crystals”. Are the authors referring to the SeMet containing crystals? There does not appear to be any other mention of metal-labeled crystals described in the main text.

Figures

- Figure 1, Panel B. The authors use a PAM with 7 significant figures for Panel A and then switch to 5 significant figures for Panel B. Consider using a consistent number of nucleotides to describe the PAM or label the nucleotide numbers on Panel B for clarity or describe what the nucleotide numbering the Figure legend.
- Figure 1, Panel C. The lanes are likely mislabeled. There is no substrate 4mC indicated for any lane.
- Figure 5 – The contrast and dimensions of the panels do not match. Where possible, the authors should very clearly define where one gel ends and another begins for splicing two gels together.
- Figure 6A – The representation of this could be improved to better highlight the differences between CC and TC PAM structures.

Reviewer #2 (Remarks to the Author):

Cas9 is an RNA guided DNA endonuclease that cleaves target double-stranded DNA via RuvC and HNH domains, and it has been utilized in many gene editing and diagnostic applications. Protospacer adjacent motif (PAM) is critical for DNA target recognition and unwinding by Cas9.

In this manuscript, the authors determined two crystal structures of AceCas9-sgRNA-DNA complex, with the correct and an incorrect PAM. The authors also performed in vivo cell survival assay and in vitro cleavage assay, and found that AceCas9 is sensitive to the methylation of the first but not the second cytosine base of PAM. This manuscript gives us detail information of AceCas9-sgRNA and AceCas9-sgRNA recognition. While there are some considerations that could improve the manuscript further.

Main points,

1. Authors described interactions between AceCas9 and Connect loop 1 (CL1) and Connect loop 2 (CL2), structural figures are needed for better displaying the interactions.
2. In Figure 4C, authors performed in vivo cell survival assay, why the numbers of clones containing AceCas9 mutants are much less than that of WT without arabinose?
3. In Fig 5, left panel, the authors test the cleavage with the DNA containing the PAM sequence "ACAAC" or "CACCC". R1088A mutant has a higher cleavage activity for the supercoiled plasmid containing the PAM "ACAAC" than "CACCC". However, the NNNCC is the best PAM sequence. In addition, why not use DNA containing the identical sequence of the first three N?
4. The authors claimed that R1088A had some activity on the substrates associated with C4* variants even when the wild-type did not (Figure 5 & Figure S5). This conclusion cannot be made from these two figures, the activity of WT AceCas9 is higher than that of R1088A in the cleavage assays in Figure 5 and Figure S5. By the way, Figure 5 is not clear enough, a high resolution one is needed.
5. The R1088A mutant has limited effect on the cleavage as shown in fig. 5, implying that the interaction between r1088 and G(-5) is not crucial. But in fig. 1B, the 5th nucleotide within the PAM almost abolished the DNA cleavage activity. Please explain the inconsistency.

6. In fig. 1B, the DNA substrate containing the PAM "ACAAC" has been completely cleaved, and DNA containing the PAM "ACATC" is half cleaved. In these two PAM sequences, the C4 has been replaced by A or T, respectively. These data suggest the mutation of C4 is tolerated and the interactions between Cas9 and C4 are not essential. However, the methylation of C4 almost abolished the DNA cleavage, suggesting the base contacts with C4 are essential. Please explain this inconsistency.

7. The authors solved a structure of Cas9 in complexed with DNA target containing the 5'-NNNTC-3' PAM (C4*T) at 3.61 Å. They found this complex has a shorter b-axis than that of the wild-type complex crystal (112 Å versus 119 Å). How many residues are not traced due to 3.6 Å low resolution? If this complex has more residues disordered than the WT complex, it cannot make the conclusion that the structure has changed. The superposition of these two complexes is suggested.

8. The last sentence of the result "Thus, the C4*T structure likely represents an inactive conformation of AceCas9-DHNH when it encounters a non-PAM DNA sequence." The observation claimed above cannot make this conclusion. In addition, the wild type complex is likely in the inactive state based on the structures of Cas9 published.

Minor points,

1. In Figure 1B, over 50% pre-linearized substrate containing "ACAAC" PAM was cleaved by AceCas9, while few was cleaved in Figure 5 and Figure S5 when using the same substrate and WT AceCas9. The authors should give an explanation or repeat these cleavage assays carefully.
2. In the caption of Figure 5, "Each DNA substrate at 100 nM concentration ...", while the DNA concentration is about 6 nM in the method. The authors should check it carefully.
3. In Figure 1D, the label "AGG4mCC" should be "AGG5mCC". The labels "linearized" and "supercoil" point at the wrong position.
4. Page 14, in the sentence "The C4*T (TT PAM) crystal diffracted to 3.61 Å", "TT PAM" should be "TC PAM".
5. In the caption of Figure 4C, "four PAM-interaction mutants" should be "three".

Reviewer #3 (Remarks to the Author):

Summary:

In this work, the authors build on their previous study by more fully characterizing the PAM specificity of *Acidothermus cellulolyticus* Cas9. Initially, they use a cleavage depletion screen to validate that 5'-NNNCC-3' is essentially the only specific PAM recognized by this Cas9 variant. Interestingly, this PAM is unique from classical PAMs in that it contains recognition of pyrimidine bases (as opposed to the classical *S. pyogenes* NGG). They go on to show that these cytosine bases are recognized in a partially methylation sensitive manner, with position 4 but not 5 being blocked by methylation, in vitro, in plasmids, and in bacterial cells. They then solve various crystal structures of the protein (lacking HNH) as well as a ternary complex with a guide RNA and substrate. They find that the structure of this Cas9 variant most closely resembles that of CdiCas9 (PDBid: 6JOO). Overall, the structure is generally consistent with previous structures of Cas9 with a few exceptions. They further probe structure-function interactions by mutating the resulting nucleic acid sequences and performing directed evolution on the PI subdomain of AceCas9 (residues 1025-1106), and solving corresponding crystal structures.

Overall, the paper is well-written, logical, and the results support the assertions made. There are also interesting structural insights as well as possible direct applicability to methylation detection. As a result, the paper has the potential to influence the field. The paper is suitable for publication in Nature

Communications pending the following revisions:

Major Points:

-In addition to the crystal structure work, the strength of the manuscript would be bolstered by adding in some binding data (Kd) to better characterize the effect of the methylated PAMs. Clearly, these are not cleaved, but the descriptions derived from this in the crystal structure do not provide data to clearly differentiate between weakened binding to the substrate or misalignment. Performing EMSAs and providing Kd values would shed light on this. In addition, it would be interesting to see how the binding affinity compares to classical Spy Cas9.

-While the authors examine methylation at C5 in a bacteria model, they do not do so for the C4 position, which is more interesting since ostensibly this is the one which is sensitive to methylation. The experiment should be performed, or a suitable explanation for why it can't be performed via expression of HaeIII should be provided.

-The consensus for bacterial methylation vs methylation in mammalian cells is quite distinct, as noted briefly in the discussion. However, the authors should comment on the applicability of this to mammalian non-CpG methylation. Methylation at CC (which matches the PAM) rather than classical CpG has been reported in mammalian cells. Perhaps an in vitro demonstration on a relevant locus in mammalian cells would also go to furthering this application.

Minor Points:

-the direct contacts between acCas9 and its guide target should be discussed in more detail. This could have important implication for specificity, as well as flexibility in terms of using chemically-modified guide RNAs (see Rueda et al., Nature Comm.; Cromwell et al., Nature Comm.)

-Throughout the paper, apostrophes (') are used in place of prime (′), e.g., 5'-NNNCC-3'

-AceCas9-DHNN should be changed to AceCas9-ΔHNN

Reviewer #4 (Remarks to the Author):

Das and colleagues report on the Type II-C CRISPR-Cas9 nuclease from *Acidothermus cellulolyticus* (AceCas9), supplementing their prior studies with structural data to explain its specificity for the cognate 5'-NNNCC-3' PAM. They identify three residues, Glu1044/Arg1088/Arg1091, that play key roles in binding to the CC dinucleotide and impart its specificity. They go on to show that AceCas9 is sensitive to the methylation status at C4 of the PAM, making it the first CRISPR nuclease to distinguish between epigenetic modifications of the targeted sequence. Overall, the work is significant given the interesting specificity of the nuclease for unmethylated DNA, where it may have applications in the detection or manipulation of epigenetic modifications. The authors unfortunately do not explore any such applications. Furthermore, significant claims are made regarding the preference of AceCas9 or its cognate gRNA for key amino acid or nucleotide identities that cannot be substantiated with the provided data, which limit my enthusiasm for those elements of this work.

Major issues:

The AceCas9 PAM was described by the author's lab as early as 2017 (*ACS Synth. Biol.* 2017, 6, 6, 1103–1113) and later revalidated in 2019 (*Methods Enzymol.* 2019, 616, 265–288). It is therefore unclear why an additional PAM depletion assay was needed in this report to further characterize the sequence determinants for binding to this motif.

Claims relating to nucleotide or amino acid preference, or lack thereof, following directed evolution are difficult to substantiate in the absence of comprehensive coverage of all possible variants. Generally speaking, deep saturation mutagenesis studies are more suited to accurately quantify preference as they purposefully generate each variant. EpPCR is more suited to yielding variants with altered or improved activities, as it can give rise to sequences encoding more than one substitution. As an example, I consider the data presented using site directed mutagenesis of Arg1088 and Arg1091. In Figures 4B,C, the authors clearly show a requirement for arginine at both positions, which agrees with other PAM interacting amino acids for homologous Cas9 variants. Results presented in the NGS data (Table 4) may be interpreted opposite to these findings, suggesting that neither amino acid is required as evidenced by limited enrichment of any amino acid at these positions. Additional issues with this analysis can be seen in considering Asp1025-Gly1042, where a stretch of 18 amino acids shows that the most preferred residue is glycine in each case, and Thr1066-Ile1106, where no single amino acid is preferred with the exception at position Leu1075. These collective points suggest that any analysis of degree of preference cannot be accurately made. These same arguments hold the gRNA mutational analysis. Since EpPCR often introduces more than one mutation per sequence, it is certainly possible that unpredictable epistatic interactions may occur, which would be occluded by calculations that rely on nucleotide abundance at a specific position. An additional issue with toxin-based selection is that plasmids may spontaneously mutate to disrupt toxin function, which would not be obvious in this experimental pipeline.

Minor issues:

Figure 4B,C: identities of mutations change between experiments (e.g. R1091K vs R1091A) or not mentioned (e.g. R1088/R1091).

Significant differences in figure size and image quality make independent analysis of the findings difficult.

Why was the HNH domain deleted prior to crystallization? Why not use a mutation to inactivate the HNH domain? If size was a constraint for crystallization, please also indicate this point in the manuscript for clarity.

Reviewer #1

Das et al., describe the recognition of an NNNCC PAM by AceCas9. Using biochemistry, the authors demonstrate that AceCas9 is sensitive to the methylation status at the C4 position within the PAM. The authors then describe the overall structure of AceCas9, the molecular details around its sgRNA recognition and the molecular basis for discriminating methylation at the C4 position. The biochemistry and structural biology are clear, and the conclusions fitting given the data, however, the manuscript would benefit greatly from improved clarity in sections and elaboration on important points as described below. Overall, the manuscript is thorough, and the data reasonably well presented throughout to give a narrative that would be of broad interest to the community provided the authors substantiate some claims.

Major Comments

1] A major focus of the manuscript is the described molecular details of the AceCas9 interaction with the NNNCCNN PAM (Figure 4). The authors adequately describe the interactions in the main text and Figure 4A and 4B are very clear. However, given the importance of the observed interactions to the narrative, it is critical that the authors include either a main text or supplementary figure panel demonstrating the quality of the electron density that supported modelling in this region. Specifically, the authors should use an unbiased map suggest as an omit map that excludes the region of interest described in Figure 4A. Depending on the clarity, the authors could include this overlaid in the boxed-out region of 4A or as an additional panel in a supplementary figure.

We have now updated Figure 4 with an omit-map focusing on the PAM region without losing the depiction of close contacts. Thanks for this suggestion for making the figure more effective.

2] Figures 1 – 4 are well presented and described in the main text. However, the manuscript would benefit substantially from improvements to both the clarity of the text and the figures surrounding the data in Figure 5. Specifically, Figure 5 is referenced in a single sentence within the main text. However, there is a lot of data presented and it needs to be clearly described.

Thanks for the reviewer to point this out. An entire paragraph is now added in the section “Recognition of the 5’-NNNCC-3’ PAM” that discusses all the results in Figure 5. In addition, we have significantly improve Figure 5 that now includes a clear summary of the results.

Additionally, the gels appear to be spliced together and formatted differently which is concerning. If supported by the journal, the authors are encouraged to include uncropped images of all gels described in the manuscript for transparency.

The spliced images were simply because of a mismatch in loading order between supercoiled and pre-linearized, not because of piecing together unrelated reactions. We actually did not need to put sets of reactions together and thus now separated them as individual panels. The original gel images including both supercoil and prelinearized are all included in Figure S5.

3] It is the opinion of this reviewer that the TC-PAM containing structure is incorrectly described as an inactive state and would be more correctly described as an artefactually active state. From a purely biophysical perspective, the sole purpose of Cas9 recognition of a PAM is to provide a nuanced distortion to the otherwise B-form helical DNA that allows for strand invasion by the pre-ordered guide RNA spacer resulting in a stable R-loop structure. When the PAM is not consensus, strand invasion is not favored regardless of the spacer sequence and thus Cas9 does not form an R-loop and cut the dsDNA. To capture the TC-PAM structure, the authors used a DNA substrate that comprises a target strand containing the target nucleotides complementary to the guide RNA spacer in addition to the TC-PAM proximal nucleotides. In contrast, the non-target strand is short and only comprises the TC-PAM and its proximal sequence. In effect, the authors have provided an already unwound target strand that is completely accessible to the sgRNA spacer bound to AceCas9. It is therefore unsurprising that the AceCas9 can bind to the target strand and form a stable structure regardless of the PAM sequence. However, as the authors noted, the PAM proximal duplex is not tightly associated with the PI domain due to the lack of the consensus 'C' at position 4 of the PAM which is itself interesting to observe. However, the implications of the substrates used, and the observed lack of PAM interaction, may be lost on some readers. The authors would benefit from describing that the mutated PAM and lack of interaction would likely prevent R-loop formation in the first place and that the only reason they observe this state is because of the substrates used.

We completely agree with the reviewer on this point and how Cas9 uses PAM to unwind the DNA duplex. We did not provide a good description of the structure on the TC PAM complex. We now add the following passages to the section describing TC PAM:

“While a non-PAM sequence in a dsDNA likely prevents unwinding and subsequent formation of the R-loop, the artificially constructed C4*T oligos lacking the paired protospacer helix in our cocrystallization studies are attached to AceCas9 RNP largely via the guide-TS heteroduplex, which allowed us to examine how the TC PAM may interact with AceCas9 PID.” “We infer that should AceCas9 encounters a dsDNA with the TC PAM, it would have a weak interaction with TC, thus preventing local unwinding and cleavage of the dsDNA substrate.”

Finally, the Figure 6A would benefit from some other features to better illustrate the distances between protein and nucleic acid. At present, it is not immediately obvious that the DNA in the TC-PAM structure is that much further from the AceCas9 PI domain.

We improved Figure 6 in two ways: 1) we now describe more clearly how the displayed protein residues were identified in Figure 6A (They were those within 3.6 Å away from the four PAM nucleotides). This should make clear to readers that fewer numbers of residues are within 3.6 Å of the TC PAM than the CC PAM; 2) we now add a ribbon representation of the DNA in the TC PAM structure that makes comparison of movement easier. Thanks for this suggestion for making the figure more effective.

4] The authors comment on how the sensitivity of AceCas9 to methylation at the C4 position presents opportunities for technology development for detecting epigenetic changes. However, as the authors describe, 70-80% of methylated cytosines occur at CpG sites. Based on the authors data, AceCas9 would not be able to discriminate this particular methylation site by

PAM recognition since it does not match the consensus. The authors should comment on this and provide some elaboration on the potential utility of the PAM sensitivity to methylation marks in the discussion especially given how much emphasis is placed on this activity.

We now added more detailed discussion in the Discussion section to include the fact that although most 5-methyl cytosine are found in CpG islands in normal mammalian tissues, methylation of non-CpG including CpC is also significant in plant cells and stem cells. We also suggest the possibility that AceCas9 can be engineered to have 5mCpG sensitivity.

5] The gap between R_{work} and R_{free} is quite large (~5%). Normally this would suggest some amount of model bias if the phases were estimated by molecular replacement alone, however, the authors used both molecular replacement and experimentally determined phase information from SeMet-crystals. Could the authors comment on the large gap? Given the methods used, it suggests some degree of overfitting has occurred. Furthermore, the authors indicate on Page 8 in the results section that the Rfactors are 0.24/0.27 (R/R_{free}), which does not match the Table or the provided deposition files.

We have now improved the gap to be ~4%, which is quite reasonable for the structure at this resolution (see the updated validation report). X-ray crystallography has inherent model biases because of the lack of measured phases. High resolutions offer some help with increased degree of freedoms. Unfortunately, this structure is at a medium resolution where we had to work hard to minimize the model biases.

We also fixed the inconsistency in reported R factors.

Minor comments

Page 5 – The sentence “Here we further characterized the PAM recognized by AceCas9 and report crystal structures of a functional AceCas9 variant bound with its guide RNA and a DNA target” is somewhat misleading. The function of Cas9 is to bind and cut dsDNA. The variants crystallized are lacking the HNH domain and are therefore non-functional. This should be reworded to clearly describe what is meant by ‘functional’.

We revised the statement to “Here we further characterized the PAM recognized by AceCas9 and report crystal structures of AceCas9 without its HNH domain (AceCas9-ΔHNH) bound with a guide RNA and DNA targets.”

Page 5 – The sentence “Mutation of the 5'-NNNCC-3' to 5'-NNNTC-3' severely impaired AceCas9 activity and caused detachment of AceCas9 from the PAM” is somewhat misleading. Detachment implies the active release of something that was once bound to AceCas9 after the introduction of the C-T change. Technically, the C-T change likely prevented stable binding in the first place rather than causing detachment. This should be reworded to better reflect the experiments and observations in the structure

We revised the statement to “Mutation of the 5'-NNNCC-3' to 5'-NNNTC-3' severely impaired AceCas9 activity and destabilized the AceCas9-PAM interaction.”

Page 5 – The sentence “The cleavage products by a functional AceCas9 or that lacking the guide RNA of a plasmid library containing a cognate protospacer followed by seven randomized base pairs were sequenced and analysed (Figure 1A)” is confusing. Consider rewording to make it clearer.

We revised the statement to “A plasmid library containing a cognate protospacer followed by seven randomized base pairs was subjected to cleavage by either a functional AceCas9 or AceCas9 without its guide RNA. The uncleaved products from both reactions were sequenced by NGS and compared to reveal the preferred PAM sequences for AceCas9 (Figure 1A).”

Page 6 – The sentence “We constructed eight plasmid DNA substrates containing various PAM sequences and subjected them to cleavage by AceCa9 both in their supercoil and pre-linearized forms” would benefit from the addition of ‘in vitro’ to make it clear that the assay uses purified proteins and nucleic acid substrates.

Fixed.

Page 9 – The sentence “The regions identified in the AceCas9 structure serve as excellent guides to protein engineering efforts in improving AceCas9 activity” seems out of place. Could the authors elaborate on what this means?

We removed this statement.

Page 13 – The authors describe a “protein directed evolution assay” however based on the data it is more accurate to describe this as a selection since there is no real ‘evolution’ happening here, just a selection for functional variants. The authors should describe it as something to the effect of a ‘functional AceCas9 selection where the input library contained 105 AceCas9 PI variants.’

The reviewer is correct that we did not “evolve” the protein. We rephrased the assay to “functional selection assay”.

Page 13 – The description of the data from Table S4 where the authors state that certain positions have preferences is confusing. Do the authors mean to say that the residues at these positions are tolerated in the described functional assay? Please consider rewording this section to improve clarity.

Description of the library screening data associated with Table S4 and the new Table S5 is significantly improved. Language was added in the text to help readers to interpret the results. In addition, Figure 4D is added for clear visualization. A related result presented in Figure 3A is also updated with more clear presentation.

Page 16 – The sentence starting “Despite low sequence homology...”. Proteins being homologous is binary. They are either homologous or they are not. The correct term to use in this sentence would be low sequence “identity”. Sequence identity is used to define if something is homologous.

Thank you. We revised the word to “identity”.

Page 19 – Sentence “For each of the heavy metal-labeled protein crystals”. Are the authors referring to the SeMet containing crystals? There does not appear to be any other mention of metal-labeled crystals described in the main text.

The statement is revised to “For each of the heavy metal-labeled crystals, whether used or not in final phase determination, x-ray wavelength was tuned....”.

We did collect Pt, Hg, Au atom labeled data but did not use these in final phasing and thus not included in description.

- Figure 1, Panel B. The authors use a PAM with 7 significant figures for Panel A and then switch to 5 significant figures for Panel B. Consider using a consistent number of nucleotides to describe the PAM or label the nucleotide numbers on Panel B for clarity or describe what the nucleotide numbering the Figure legend

This is a misunderstanding. We included seven randomized nucleotides in our PAM library for completeness so we could analyze the PAM requirement for all seven nucleotides. We did discovery that positions labeled as “N” do not confer specificity in the plasmid depletion assay (see Table S1). Thus, we kept the first five nucleotides in in vitro cleavage experiments in Figure 1B. But we see that the 7mer labeled “PAM” in Figure 1A could be misleading and thus changed it to “NNNCC”. We also included a statement in Figure 1 caption that “Seven base pairs immediately downstream of the protospacer are randomized.”.

Figure 1, Panel C. The lanes are likely mislabeled. There is no substrate 4mC indicated for any lane.

Thank you. We fixed the error.

Figure 5 – The contrast and dimensions of the panels do not match. Where possible, the authors should very clearly define where one gel ends and another begins for splicing two gels together.

We re-made Figure 5 completely to avoid the artifacts.

- Figure 6A – The representation of this could be improved to better highlight the differences between CC and TC PAM structures.

This figure is now improved with better display to reveal the difference between the two structures. In addition, the figure caption is revised to aid understanding of the images.

Reviewer #2

Cas9 is an RNA guided DNA endonuclease that cleaves target double-stranded DNA via RuvC and HNH domains, and it has been utilized in many gene editing and diagnostic applications. Protospacer adjacent motif (PAM) is critical for DNA target recognition and unwinding by Cas9.

In this manuscript, the authors determined two crystal structures of AceCas9-sgRNA-DNA complex, with the correct and an incorrect PAM. The authors also performed in vivo cell survival assay and in vitro cleavage assay, and found that AceCas9 is sensitive to the methylation of the first but not the second cytosine base of PAM. This manuscript gives us detail information of AceCas9-sgRNA and AceCas9-sgRNA recognition. While there are some considerations that could improve the manuscript further.

1. Authors described interactions between AceCas9 and Connect loop 1 (CL1) and Connect loop 2 (CL2), structural figures are needed for better displaying the interactions.

Three new panels are now added to Figure 3 that display detailed CL1-AceCas9 and CL2-AceCas9 interactions.

2. In Figure 4C, authors performed in vivo cell survival assay, why the numbers of clones containing AceCas9 mutants are much less than that of WT without arabinose?

The difference in colony numbers on arabinose-minus (or +chloramphenicol) plates between any two transformations is simply the result of transformation efficiencies of the plasmids. However, we only need to compare the number of colonies between the +arabinose and +chloramphenicol plates of the same plasmid, which is not impacted by the different efficiency of different plasmids.

3. In Fig 5, left panel, the authors test the cleavage with the DNA containing the PAM sequence "ACAAC" or "CACCC". R1088A mutant has a higher cleavage activity for the supercoiled plasmid containing the PAM "ACAAC" than "CACCC. However, the NNNCC is the best PAM sequence. In addition, why not use DNA containing the identical sequence of the first three N?

The reviewer is correct that R1088A cleaved supercoiled ACAAC slightly better than cleaving CACCC. However, it still cleaved pre-linearized CACCC better than ACAAC, suggesting it may have a different dependence on DNA super-helicity. Note that the wild-type enzyme behaved as expected to favor NNNCC as the PAM. This data does not contradict to the established NNNCC PAM for the wild-type but revealed an interesting behavior of the R1088A mutant enzyme in favoring a different PAM or pre-linearized DNA, which may help us engineer AceCas9 for an alternative PAM in the future.

The reason we varied the first three Ns is to confirm our plasmid depletion analysis in which the first three Ns did not show effects on PAM recognition in cellular assays (Figure 1A), although there are slight differences in cleavage activities, say between TGTCC and GTGCC. These results indicate that in vitro biochemistry does not necessarily reflect in vivo activities.

4. The authors claimed that R1088A had some activity on the substrates associated with C4* variants even when the wild-type did not (Figure 5 & Figure S5). This conclusion cannot be

made from these two figures, the activity of WT AceCas9 is higher than that of R1088A in the cleavage assays in Figure 5 and Figure S5. By the way, Figure 5 is not clear enough, a high resolution one is needed.

The reviewer is correct. We now removed this statement.

Figure 5 is now significantly improved.

5. The R1088A mutant has limited effect on the cleavage as shown in fig. 5, implying that the interaction between r1088 and G(-5) is not crucial. But in fig. 1B, the 5th nucleotide within the PAM almost abolished the DNA cleavage activity. Please explain the inconsistency.

Altering PAM sequences has different consequences than altering PAM-interacting residues. Mutation of a PAM nucleotide in fact means a change of two nucleotides, one on the target strand and one on the nontarget strand. Though R1088 does not directly contact the “C” at the 5th position, it contacts its complementary “G”. Mutation of “C” to any other nucleotides would change its complementary “G” as well, thus impacting R1088 interactions. Mutation of R1088 to alanine removes its polar interaction with the complementary “G”. Thus, a mutation of PAM nucleotide is more deleterious than mutation of R1088.

6. In fig. 1B, the DNA substrate containing the PAM "ACAAC" has been completely cleaved, and DNA containing the PAM "ACATC" is half cleaved. In these two PAM sequences, the C4 has been replaced by A or T, respectively. These data suggest the mutation of C4 is tolerated and the interactions between Cas9 and C4 are not essential. However, the methylation of C4 almost abolished the DNA cleavage, suggesting the base contacts with C4 are essential. Please explain this inconsistency.

One needs to consider the complementary bases in C4 mutations, not just the C to A or the C to T change. Still, how exactly AceCas9 accommodates ACAAC better than ACATC remains unexplained at this time, although neither could be cleaved if it is in a pre-linearized DNA or in cell, suggesting both are very weak PAMs, if they are in nature.

The effect of DNA methylation differs from DNA mutation in that it adds additional group to an otherwise base-paired duplex. We do not have enough biophysical data but can speculate that DNA methylation can cause changes in DNA dynamic behaviors that is detrimental to AceCas9 recognition. We would be interested in explaining the biophysical principle of this observation with more in depth studies. Our new binding experiment that compared binding constants of methylated and non-methylated DNA provided the first evidence that methylation likely impact the unwinding rather than binding step of AceCas9-DNA interaction (Figure S2).

7. The authors solved a structure of Cas9 in complexed with DNA target containing the 5'-NNNTC-3' PAM (C4*T) at 3.61 Å. They found this complex has a shorter b-axis than that of the wild-type complex crystal (112 Å versus 119 Å). How many residues are not traced due to 3.6 Å low resolution? If this complex has more residues disordered than the WT complex, it cannot make the conclusion that the structure has changed. The superposition of these two complexes is suggested.

There is a notable conformational difference as we indicated in Figure 6. The 7 Å difference in cell dimension is the result of this conformational difference between the two structures.

8. The last sentence of the result "Thus, the C4*T structure likely represents an inactive conformation of AceCas9-DHNH when it encounters a non-PAM DNA sequence." The observation claimed above cannot make this conclusion. In addition, the wild type complex is likely in the inactive state based on the structures of Cas9 published.

We agree that most crystallographic studies use artificial constructs and subject to crystal packing and thus may not truly represent "functional" or "non-functional" state. We thus should avoid these terms.

Based on the comment of reviewer 1, we made a substantial change in this section, which includes addressing the artificial construct of the TC PAM. We now conclude with what the structures actually tell us.

Minor points,

1. In Figure 1B, over 50% pre-linearized substrate containing "ACAAC" PAM was cleaved by AceCas9, while few was cleaved in Figure 5 and Figure S5 when using the same substrate and WT AceCas9. The authors should give an explanation or repeat these cleavage assays carefully

We should have explained that Figure 5 reactions were of 15 minutes while Figure 1 reactions were of 1 hour. The shorter reaction time for Figure 5 was for an attempt to observe difference in cleavage rates between the wild-type and the mutants.

2. In the caption of Figure 5, "Each DNA substrate at 100 nM concentration ...", while the DNA concentration is about 6 nM in the method. The authors should check it carefully.

Fixed.

3. In Figure 1D, the label "AGG4mCC" should be "AGG5mCC". The labels "linearized" and "supercoil" point at the wrong position.

Fixed.

4. Page 14, in the sentence "The C4*T (TT PAM) crystal diffracted to 3.61 Å", "TT PAM" should be "TC PAM".

Fixed.

5. In the caption of Figure 4C, "four PAM-interaction mutants" should be "three".

Fixed.

Reviewer #3

In this work, the authors build on their previous study by more fully characterizing the PAM specificity of *Acidothermus cellulolyticus* Cas9. Initially, they use a cleavage depletion screen to validate that 5'-NNNCC-3' is essentially the only specific PAM recognized by this Cas9 variant. Interestingly, this PAM is unique from classical PAMs in that it contains recognition of pyrimidine bases (as opposed to the classical *S. pyogenes* NGG). They go on to show that these cytosine bases are recognized in a partially methylation sensitive manner, with position 4 but not 5 being blocked by methylation, in vitro, in plasmids, and in bacterial cells. They then solve various crystal structures of the protein (lacking HNH) as well as a ternary complex with a guide RNA and substrate. They find that the structure of this Cas9 variant most closely resembles that of CdiCas9 (PDBid: 6J00). Overall, the structure is generally consistent with previous structures of Cas9 with a few exceptions.

They further probe structure-function interactions by mutating the resulting nucleic acid sequences and performing directed evolution on the PI subdomain of AceCas9 (residues 1025-1106), and solving corresponding crystal structures.

Overall, the paper is well-written, logical, and the results support the assertions made. There are also interesting structural insights as well as possible direct applicability to methylation detection. As a result, the paper has the potential to influence the field. The paper is suitable for publication in Nature Communications pending the following revisions:

-In addition to the crystal structure work, the strength of the manuscript would be bolstered by adding in some binding data (Kd) to better characterize the effect of the methylated PAMs. Clearly, these are not cleaved, but the descriptions derived from this in the crystal structure do not provide data to clearly differentiate between weakened binding to the substrate or misalignment. Performing EMSAs and providing Kd values would shed light on this. In addition, it would be interesting to see how the binding affinity compares to classical Spy Cas9.

This is a very good suggestion. We designed a competition assay, which allowed us to estimate binding constants of oligo substrates that are no methylation, 5mC4* methylated or 5mC5* methylated (Figure S2B). We concluded that methylation only minimally impacts AceCas9 binding to DNA, thus likely the unwinding step. We added this data to Figure S2. The result was incorporated into the Result section.

While the authors examine methylation at C5 in a bacteria model, they do not do so for the C4 position, which is more interesting since ostensibly this is the one which is sensitive to methylation. The experiment should be performed, or a suitable explanation for why it can't be performed via expression of HaeIII should be provided.

This is a very good suggestion. We have spent quite a bit of time in trying to establish such an assay in *E. coli*. Our idea is to establish an HaeIII-expressing host where we can assay for AceCas9 activity. Although we have obtained all components, we discovered several problems, with HaeIII methyltransferase being toxic to cells the most difficult to overcome. We do have several more strategies to try that, unfortunately, will take some more time. We wonder if we can delay this experiment while reviewers are re-evaluating the rest of the revisions and include this experiment in the next round, if necessary.

Throughout the paper, apostrophes (') are used in place of prime (′), e.g., 5′-NNNCC-3′

Fixed.

AceCas9-DHNNH should be changed to AceCas9-ΔHNNH

Fixed.

Reviewer #4

Das and colleagues report on the Type II-C CRISPR-Cas9 nuclease from *Acidothermus cellulolyticus* (AceCas9), supplementing their prior studies with structural data to explain its specificity for the cognate 5′-NNNCC-3′ PAM. They identify three residues, Glu1044/Arg1088/Arg1091, that play key roles in binding to the CC dinucleotide and impart its specificity. They go on to show that AceCas9 is sensitive to the methylation status at C4 of the PAM, making it the first CRISPR nuclease to distinguish between epigenetic modifications of the targeted sequence. Overall, the work is significant given the interesting specificity of the nuclease for unmethylated DNA, where it may have applications in the detection or manipulation of epigenetic modifications.

The authors unfortunately do not explore any such applications. Furthermore, significant claims are made regarding the preference of AceCas9 or its cognate gRNA for key amino acid or nucleotide identities that cannot be substantiated with the provided data, which limit my enthusiasm for those elements of this work.

In the revised Discussion section, we discuss explicitly some possible applications of DNA methylation sensitivity by AceCas9.

Major issues:

The AceCas9 PAM was described by the author's lab as early as 2017 (ACS Synth. Biol. 2017, 6, 6, 1103–1113) and later revalidated in 2019 (Methods Enzymol. 2019, 616, 265–288). It is therefore unclear why an additional PAM depletion assay was needed in this report to further characterize the sequence determinants for binding to this motif.

Neither of the earlier works (ACS Synth. Biol. 2017, 6, 6, 1103–1113 or Methods Enzymol. 2019, 616, 265–288) was done via a library-based method. As a result, although they provided the most significant PAM sequence and enabled studies of other properties of AceCas9, they could not eliminate other possible PAMs. In this work, we performed PAM identification from ground up by using a complete 7-mer PAM library followed by NextGen sequencing. This is the correct method for identify PAM for a new CRISPR-Cas enzyme.

Claims relating to nucleotide or amino acid preference, or lack thereof, following directed evolution are difficult to substantiate in the absence of comprehensive coverage of all possible variants. Generally speaking, deep saturation mutagenesis studies are more suited to

accurately quantify preference as they purposefully generate each variant. EpPCR is more suited to yielding variants with altered or improved activities, as it can give rise to sequences encoding more than one substitution. As an example, I consider the data presented using site directed mutagenesis of Arg1088 and Arg1091. In Figures 4B,C, the authors clearly show a requirement for arginine at both positions, which agrees with other PAM interacting amino acids for homologous Cas9 variants. Results presented in the NGS data (Table 4) may be interpreted opposite to these findings, suggesting that neither amino acid is required as evidenced by limited enrichment of any amino acid at these positions.

The reviewer clearly understands the method well and correctly pointed out the limitation of this method. The diversity of starting library was limited by the available error-prone PCR method we used. Most positions usually contain over 96% the wild-type nucleotide or amino acid and only a few percent (or zero) of other 19 amino acids. Nonetheless, our careful analysis of only non-wild-type sequences did provide valuable data in both the RNA and PID screens as described below:

1. With regard to Table S4 (not Table 4), the original version included some artifacts that our in-house analysis program failed to remove. We discovered that some sequences contain stretches of poly-glycine. Although we still do not understand their origin, the revised code cleaned up this artifact and tabulated the entire collection of protein sequences before the frequency analysis. The revised result is much more accurate, although did not change the conclusion qualitatively;
2. With regard to consistency between Table S4 and Figure 4. The data do generally agree. For instance, R1088A did not survive in our cellular assay (Figure 4C) while R1088 was also selected against ala in the library assay with a negative difference frequency (-0.1088%, Table S4). While we cannot compare R1091K results because our library did not contain this variant to begin with, R1091 itself was highly selected for with a positive difference frequency (3.568%, Table S4) and all other countable amino acids were selected against (Table S4). In order to facilitate more detailed comparison of these key residues, we now include a new Table S5 that lists the frequency of each amino acid in both the library and survivor pools where one can see that R1091 was present with a 95.8% and 99.4% frequency in the input and survivor pool, respectively. Much stronger selection was seen for Glu1044 because it was present at a much lower frequency in the input pool (80.2%). The original Table S5 is now Table S6;
3. We generated two new figure panels that better display the screening results: Figure 3A for RNA screening and Figure 4D for the PID screening. The difference frequency plots are superimposed with secondary structure features of the RNA or the protein so readers can quickly glean the significance. The trend of conservation in both RNA and PID is in fact consistent with our structural data. We also observed co-variations of the base pairs (for instance, 52:66, 79:98, and 80:97);
4. Finally, we now added overall statistical data that can help readers understand the extend of variations in both pools. For instance, while the total number of sequences are similar in the RNA library and survivor pools, those of unique sequences differ drastically with survivor pool is only 9% of that of the library pool (Table S3). This

suggests wild-type is strongly selected for in our screening assay. It is thus quite conserved.

Additional issues with this analysis can be seen in considering Asp1025-Gly1042, where a stretch of 18 amino acids shows that the most preferred residue is glycine in each case, and Thr1066-Ile1106, where no single amino acid is preferred with the exception at position Leu1075. These collective points suggest that any analysis of degree of preference cannot be accurately made. These same arguments hold the gRNA mutational analysis.

As described above, we revised the in-house program and removed the poly-glycine artifact. Leu1075 is in fact not preferred because it has a negative difference frequency (less abundant in survivor pool than in the library pool). In fact, position 1075 strongly prefers valine instead of leucine (Table S4 and Table S5). We believe that this kind of results demonstrates the power of the screening method, which can be followed up with biochemical analysis. Table S5 lists more detailed comparison between the library and survivor pools. Note valine was present at this position with a 76.8% frequency in the entire survivor pool.

Since EpPCR often introduces more than one mutation per sequence, it is certainly possible that unpredictable epistatic interactions may occur, which would be occluded by calculations that rely on nucleotide abundance at a specific position.

The reviewer is correct that we did not directly observe epistatic interactions. This is a result of two experimental limitations. For the PID library, two sets of independent primers had to be used in NextGen sequencing, making the two halves uncoupled. Second, the error-prone PCR method had a low mutation rate, which led to a dominate single-amino acid mutations (over 60% in libraries). Thus, we believe that we should focus on analysis of amino acid frequencies.

An additional issue with toxin-based selection is that plasmids may spontaneously mutate to disrupt toxin function, which would not be obvious in this experimental pipeline.

We believe that the chance for either the AceCas9 plasmid or the ccdB-encoding plasmid to disrupt ccdB function is very low. We perform this assay quite often and have not yet observed such phenomenon. In our new analysis, we listed every single sequence that went into the frequency calculation, which would preclude artifacts such as incomplete protein sequences.

Figure 4B,C: identities of mutations change between experiments (e.g. R1091K vs R1091A) or not mentioned (e.g. R1088/R1091).

Fixed.

Significant differences in figure size and image quality make independent analysis of the findings difficult.

All figures are now in letter size.

Why was the HNH domain deleted prior to crystallization? Why not use a mutation to inactivate the HNH domain? If size was a constraint for crystallization, please also indicate this point in the manuscript for clarity

We would have loved crystal structures of a full-length AceCas9. It did not crystallize under the conditions we tried. Thus, the use of HNH deletion is sheerly due to its successful crystallization. Despite the absence of the HNH domain, the current structure clearly revealed many features of AceCas9 critical to its function. We now added a statement to reflect this limitation.

REVIEWER COMMENTS

Reviewer #1 (Remarks to the Author):

The authors have addressed my scientific concerns and significantly improved the clarity of the narrative post-revision. The ability of AceCas9 to sense the nucleotide content of the target strand proximal region is interesting and will be of relative interest to users of CRISPR-Cas technologies. The manuscript would be strengthened by an example of AceCas9 application to detecting methylation marks in vivo, however this is likely beyond the scope of the study in question.

With some minor corrections, the manuscript is suitable for publication in Nature Communications.

Minor Comments

- Page 16 – The sentence “It is remarkable that, despite a low sequence homology...” should read “It is remarkable that, despite a low sequence identity...”

Reviewer #2 (Remarks to the Author):

In the revised version, the authors performed new assays, rewrote the manuscript for clarification, and answered all reviewers' questions. The revised paper is much better than the original version, and is suitable for publication in Nature Communications.

Reviewer #3 (Remarks to the Author):

The authors have adequately addressed my concerns.

Reviewer #4 (Remarks to the Author):

Thank you to the authors for their revised manuscript. While the quality of the data and presentation have been improved, I nonetheless continue to have major reservations about the quality of the selection pipelines and the interpretation of the results. I also echo the sentiment of Reviewer #1 with regards to the “evolution” of these genome editing components. Whereas the sgRNA and Cas9 were subjected to mutagenesis and functional selection, the findings of possible impacts of specific mutations were rarely explored beyond the PAM depletion assay.

I believe that description of a methylation-sensitive Cas9 homolog would be a useful addition to the genome editing repertoire, but I find that the selection methodologies and their interpretation are not

Major comments:

1. Regarding PAM studies, both prior reports used the established library-based methods to identify functional PAM sites, counter to the authors statement that “neither of the earlier works...was done via a library-based method”. The innovation here then is the use of NextGen sequencing to identify putative PAMs with greater depth, which is not a significant enough advance in my opinion.

Excerpt from ACS Synth. Biol. 2017:

“We designed and constructed a DNA library bearing random sequences in the PAM region as the substrates for AceCas9 cleavage (Figure S1). The PAM region contains seven randomized base pairs (5'-NNNNNNN-3' with a 25% probability for each base pair at any given position) and is located downstream of a 20-nt protospacer inserted into the pUC19 plasmid (Figure S1A).”

Excerpt from Methods Enzymol. 2019:

"We constructed a PAM library containing seven randomized nucleotides at the 3' end of the protospacer incorporated onto pUC19 (5'-NNNNNNN-3') and ~500bps away from a downstream BamHI site (Fig. 2)."

2. With respect to the glycine artifact in the sequencing data, it is highly concerning that such a large artifact of unknown origin was recognized by the authors but retained without discussion in the initial manuscript submission. With this in mind, the quality of the remaining data is difficult to determine, and the findings even after correction still do not agree with one another.

Position R1088:

- Figure 4B: R1088A is nearly as active as wildtype
- Figure 4C: R1088A is non-functional in cells
- Table S4: the greatest degree of enrichment was 1.23% (Arg) and the greatest degree of depletion was 0.89% (Gln). Alanine was depleted by 0.10%.

Position R1091:

- Figure 4B: R1091K is less active than wildtype
- Figure 4C: R1091A is non-functional in cells
- Table S4: the greatest degree of enrichment was 3.57% (Arg) and the greatest degree of depletion was 1.54% (Trp). Alanine was depleted by 0.07% and lysine was not covered within the library.

Position E1044:

- Figure 4B: E1044A is less active than wildtype, but more active than R1091K
- Figure 4C: E1044A is non-functional in cells
- Table S4: the greatest degree of enrichment was 18.1% (Glu) and the greatest degree of depletion was 14.5% (Lys). Alanine was depleted by 0.13%.

While the data presented and interpreted in Figures 4B,4C are expected to be sound, comparison with data presented in Table S4 (and used to generate Figure 4D) highlights extremely poor degrees of coverage for possible variants within the gene. Furthermore, such low magnitude changes (with exception of E1044) cannot be robustly interpreted given the significant impact of some mutations as shown using biochemical characterization. I continue to have major reservations about the quality of the mutagenized library, the library preparation for sequencing, and sequencing analysis itself that collectively influence interpretation of these findings.

3. From data presented in Table S4, positions D1025 (-4.86%), S1027 (-11.4%), T1074 (-5.56%), and L1075 (-74.7%) are highly depleted. Given the more minor effects in library enrichment/depletion observed for mutations at important positions E1044, R1088 and R1091, the authors should biochemically validate these newly discovered variants. If successful, that would lend significant confidence to the findings of the error-prone PCR-based selection approach.

4. Furthermore, data in Table S3 is presented without the wildtype base at each position, whereas Table S4 includes the abundances of all wildtype and mutant amino acids at the tested positions. Why were the wildtype nucleotides omitted? Can the authors modify Table S3 to reflect the true abundances of all bases at all positions?

Reviewer #4 (Remarks to the Author):

Thank you to the authors for their revised manuscript. While the quality of the data and presentation have been improved, I nonetheless continue to have major reservations about the quality of the selection pipelines and the interpretation of the results. I also echo the sentiment of Reviewer #1 with regards to the “evolution” of these genome editing components. Whereas the sgRNA and Cas9 were subjected to mutagenesis and functional selection, the findings of possible impacts of specific mutations were rarely explored beyond the PAM depletion assay.

I believe that description of a methylation-sensitive Cas9 homolog would be a useful addition to the genome editing repertoire, but I find that the selection methodologies and their interpretation are not

Major comments:

1. Regarding PAM studies, both prior reports used the established library-based methods to identify functional PAM sites, counter to the authors statement that “neither of the earlier works...was done via a library-based method”. The innovation here then is the use of NextGen sequencing to identify putative PAMs with greater depth, which is not a significant enough advance in my opinion.

Excerpt from ACS Synth. Biol. 2017:

“We designed and constructed a DNA library bearing random sequences in the PAM region as the substrates for AceCas9 cleavage (Figure S1). The PAM region contains seven randomized base pairs (5'-NNNNNNN-3' with a 25% probability for each base pair at any given position) and is located downstream of a 20-nt protospacer inserted into the pUC19 plasmid (Figure S1A).”

Excerpt from Methods Enzymol. 2019:

“We constructed a PAM library containing seven randomized nucleotides at the 3' end of the protospacer incorporated onto pUC19 (5'-NNNNNNN-3') and ~500bps away from a downstream BamHI site (Fig. 2).”

If we have given readers the impression that the plasmid depletion assay was innovative, we did not mean to. This is a standard method for Cas9's PAM to be thoroughly characterized. The reason we employed this method here is 1) that it has never been performed on AceCas9; and 2) to support the structural characterization of AceCas9 PAM, which was not available previously.

The library was indeed used in the previous two works but the PAM was not thoroughly identified. Both publications referred to the same experiment where we explored the cleaved fragments (~500 bps double-digested by both AceCas9 at its PAM and BamHI) rather than the uncleaved plasmids as done in this work. Previously, the cleaved fragments were cloned into a storage plasmid to generate a library of cleaved PAM. We then used Sanger sequencing method to individually sequence 18 clones that all showed the NNNCC PAM. Rather, here we used NextGen sequencing to sequence all uncleaved plasmids, which allowed us to identify all possible PAM sequences. As reviewer can see that while NNNAC is the strongest PAM sequence, NNNAC is also a weak PAM that we would not have been able to identify previously. I hope that this point is clear now and that the plasmid depletion experiment is necessary but is not the innovative part of this work.

2. With respect to the glycine artifact in the sequencing data, it is highly concerning that such a large artifact of unknown origin was recognized by the authors but retained without discussion in the initial manuscript submission. With this in mind, the quality of the remaining data is difficult to determine,

and the findings even after correction still do not agree with one another.

With regard to the presence of glycine codons in the libraries, we further consulted several experts and concluded that it is most likely sequencing artifacts. The two-color chemistry used in our MiSeq instrument would recognize weak signals as guanine in the base calling stage, which could result in a small percentage of poly guanine (which encodes glycine). Typically, commercial softwares would have them trimmed as artifacts in initial filtering steps. Our in-house program reads the raw and unfiltered sequences and thus picked up these poly guanine sequences. These should be removed as we did.

Position R1088:

- Figure 4B: R1088A is nearly as active as wildtype
- Figure 4C: R1088A is non-functional in cells
- Table S4: the greatest degree of enrichment was 1.23% (Arg) and the greatest degree of depletion was 0.89% (Gln). Alanine was depleted by 0.10%.

Position R1091:

- Figure 4B: R1091K is less active than wildtype
- Figure 4C: R1091A is non-functional in cells
- Table S4: the greatest degree of enrichment was 3.57% (Arg) and the greatest degree of depletion was 1.54% (Trp). Alanine was depleted by 0.07% and lysine was not covered within the library.

Position E1044:

- Figure 4B: E1044A is less active than wildtype, but more active than R1091K
- Figure 4C: E1044A is non-functional in cells
- Table S4: the greatest degree of enrichment was 18.1% (Glu) and the greatest degree of depletion was 14.5% (Lys). Alanine was depleted by 0.13%.

While the data presented and interpreted in Figures 4B,4C are expected to be sound, comparison with data presented in Table S4 (and used to generate Figure 4D) highlights extremely poor degrees of coverage for possible variants within the gene. Furthermore, such low magnitude changes (with exception of E1044) cannot be robustly interpreted given the significant impact of some mutations as shown using biochemical characterization. I continue to have major reservations about the quality of the mutagenized library, the library preparation for sequencing, and sequencing analysis itself that collectively influence interpretation of these findings.

We realize that the typical error-prone PCR method gives rise to low degree of variation. For this reason, we removed all wild-type sequences and computed variants among the mutants themselves in the library-based method (see response to item 4).

Results in Figure 4 and Table S4 are not in conflict. Note that Figure 4 was obtained by in vitro DNA cleavage and Table S4 was by the cell survival assay. Showing DNA cleavage activity under the single turnover condition over a long incubation time does not necessarily reflect the rate of survival in the cell-based assay. Of course, no cleavage in vitro would always correspond to no survival in cell-based assay. We believe that a certain threshold in AceCas9 cleavage rate must be met in order for it to eliminate the ccdB plasmid toxicity for cells to survive. Thus, the activities the reviewer listed do not in fact conflict and correctly indicate the essential roles of these three residues. Mutations of any one of these residues rendered inactivity of AceCas9 in cells despite some activity in vitro.

3. From data presented in Table S4, positions D1025 (-4.86%), S1027 (-11.4%), T1074 (-5.56%), and L1075 (-74.7%) are highly depleted. Given the more minor effects in library

enrichment/depletion observed for mutations at important positions E1044, R1088 and R1091, the authors should biochemically validate these newly discovered variants. If successful, that would lend significant confidence to the findings of the error-prone PCR-based selection approach.

We agree with the reviewer that some of the gain-of-function mutations seem to be interesting and the substitutions would otherwise be identified by typical structure-guided mutagenesis studies. To follow these up would be of interests to 1) understand their role in AceCas9 activity and 2) to validate the library-based functional assay. We are a bit of concerned to initiate these studies here because it would take substantial amount of efforts. We would need to include detailed kinetics analysis of each of these mutants and compare which kinetic steps they might have altered. We would also need to include a battery of different DNA substrates in order to understand how DNA substrate such as superhelicity, impacts their rates of cleavage when comparing to the wild-type enzyme. We feel that, although interesting, these studies are beyond the scope of this work here.

4. Furthermore, data in Table S3 is presented without the wildtype base at each position, whereas Table S4 includes the abundances of all wildtype and mutant amino acids at the tested positions. Why were the wildtype nucleotides omitted? Can the authors modify Table S3 to reflect the true abundances of all bases at all positions?

Table S3 and Table S4 do not conflict in presentation. We omitted full-length wild-type sequences (meaning every position contains the wild-type sequence) in both cases because they represent the LARGE MAJORITY of the sequence reads in both libraries. By omitting the full-length wild-type sequences, we could focus on the variants that retain the wild-type activity (survival). Note that despite that we omitted the full-length wild-type sequences in the analysis, all positions (nucleotide or amino acids) under consideration individually would still have high percent of wild-type sequences.

Example:

Wild-type:GDLSMF....
Variant 1 GDVSMF....
Variant 2GELSMG....

Here “Wild-type” sequence is removed while variants 1 & 2 are retained before analysis. Positions would have 100% presence of the wild-type amino acids (G, S, M).

REVIEWERS' COMMENTS

Reviewer #3 (Remarks to the Author):

The authors have addressed all experimental issues and have adequately addressed experimental comments.

Reviewer #4 (Remarks to the Author):

Thank you to the authors for their most recent responses. They have addressed my concerns with these explanations.